# A scalable gut epithelial organoid model reveals the genome-wide colonization landscape of a human-adapted pathogen

Maria Letizia Di Martino [1] ✉, Laura Jenniches [2] ✉, Anjeela Bhetwal [1], Jens Eriksson[1], Ana C. C. Lopes [1], Angelika Ntokaki [1], Martina Pasqua[1,11], Magnus Sundbom [3], Martin Skogar[3], Wilhelm Graf[3], Dominic-Luc Webb[4], Per M. Hellström[4], André Mateus [5,6], Lars Barquist [2,7,8,9] ✉ & Mikael E. Sellin [1,10] ✉

Studying the pathogenesis of human-adapted microorganisms is challenging, since small animal models often fail to recapitulate human physiology. Hence, the comprehensive genetic and regulatory circuits driving the infection process of principal human pathogens such as *Shigella flexneri* remain to be defined. We combined large-scale *Shigella* infections of enteroids and colonoids with transposon-directed insertion sequencing and Bayesian statistical modeling to address infection bottlenecks, thereby establishing the comprehensive genome-wide map of *Shigella* genes required to infect human intestinal epithelium. This revealed the *Shigella* virulence effectors essential for epithelial cell colonization across geometries and intestinal segments, identified over 100 chromosomal genes involved in the process and uncovered a post-transcriptional mechanism whereby tRNA-modification enzymes and differential codon usage exert global control of a bacterial virulence program. Our findings provide a broadly applicable framework for combining advanced organotypic tissue culture with functional genomics and computational tools to map human–microorganism interactions at scale.

*Shigella flexneri* is a principal human-/primate-restricted pathogen responsible for bacillary dysentery (reviewed in refs. 1,2). As in other human-adapted microorganisms, in vivo molecular studies addressing *Shigella* pathogenesis have been challenging due to the shortage of animal models that mimic human physiology (reviewed in refs. 3,4). Previous work has identified *Shigella* virulence factors promoting invasion, cytosolic colonization and spread within nonphagocytic cells, reliant on a type-3-secretion-system (T3SS) and cognate effectors (reviewed in refs. 5,6). Most of these virulence factors are encoded on a large virulence plasmid (pINV) and their expression is tightly controlled by environmental stimuli[7,8]. Whereas tumor-derived cell lines

have been used to map contributions of *Shigella* T3SS effectors and select chromosomal loci to invasive behavior (reviewed in refs. 9,10), these models show transformed features (for example, ref. 11), having altered morphology, metabolism, signaling and cell death pathways compared to native epithelia. Limitations of animal models have also hampered the application of in vivo high-throughput forward genetics screens. Therefore, the contribution of most of *Shigella*'s genome to colonization of a physiologically arranged gut epithelium remains undetermined.

Intestinal epithelial organoids (for example, enteroids and colonoids)—three-dimensional (3D) gut epithelial assemblies derived

A full list of affiliations appears at the end of the paper. ✉e-mail: ml.dimartino@imbim.uu.se; laura.jenniches@helmholtz-hiri.de; lars.barquist@helmholtz-hiri.de; mikael.sellin@imbim.uu.se

from stem cells[12]—have become promising tools for host–pathogen interaction studies[13–20]. *Shigella* preferentially infect intestinal epithelial cells (IECs) from the basolateral side following epithelial traversal[21]. Therefore, we developed a large-scale infection protocol for 3D-suspension-grown human enteroids with basal-out (BO) polarity[16,22] and combined this with transposon-directed insertion sequencing (TraDIS)[23]—a functional genomics approach coupling transposon mutagenesis with next-generation sequencing to link effects of gene disruptions to specific phenotypes.

A key challenge in applying genome-scale screening to an infection assay is the presence of population bottlenecks[23]. Bottlenecks occur in infection models when factors such as barrier defences, nutritional restriction or competition for colonization sites lead to transient population size reduction irrespective of the fitness of individual mutants[24]. This causes stochastic loss of mutants without a true virulence phenotype, thereby distorting results. To overcome this challenge, we systematically optimized the experimental procedures and developed a Bayesian statistical framework[25] based on the zero-inflated negative binomial (ZINB) distribution to extract mutant fitness parameters. We applied this to analyze 43 parallel enteroid bulk infections with *Shigella* Tn5 mutant sub-libraries, establishing a deep genome-wide map of *Shigella* genes required to colonize nontransformed gut epithelium. Informed by the TraDIS results, we mapped and quantified differences in colonization efficiency of *Shigella* wild-type and virulence gene mutants across cell geometries and intestinal segments, performing barcoded infections in enteroids and colonoids exposing the basal or apical surface outwards[26,27]. Finally, this work identified a post-transcriptional mechanism, linking MnmE/G U34-dependent tRNA modifications to global control of the *Shigella* T3SS virulence cascade. This study expands our understanding of enterobacterial pathogenesis and provides a generalizable framework for genome-wide forward genetics screens in complex infection models.

## Results

### A scalable human gut epithelium model for *Shigella* infection

To establish a large-scale human intestinal epithelial model for *Shigella* infection, we developed a high-throughput gentamicin protection assay, using suspension cultures of enteroids with BO polarity[16,22] (Extended Data Fig. 1a). To explore *Shigella* infection dynamics, we infected the enteroids with wild-type *Shigella flexneri* M90T harboring the intracellular reporter p*uhpT*-GFP[13,28], or a constitutive pmCherry reporter[29]. At 3 h post-infection (h.p.i.) *Shigella* infection foci appeared in IECs. Between 3 and 9 h.p.i., foci increased in size and cell-to-cell spread was also evident (Fig. 1a,b and Supplementary Videos 1 and 2).

Using TraDIS to map the *Shigella* geneset necessary for enteroid colonization requires a well-controlled experimental design. We systematically addressed the following critical parameters: (1) the size of the intracellular bacterial population, (2) the impact of infection bottlenecks and (3) the effect of bacterial growth enrichment steps following infection[23,30]. To estimate the size of the intracellular bacterial population, ~800 enteroids were infected with a mix of wild-type and Δ*mxiD* (noninvasive; lacks a T3SS structural component) strains at multiplicity of infection (MOI) 40. At 6 h.p.i., we consistently retrieved ~100,000 colony-forming units (CFUs) per infection for the wild-type *Shigella* strain and >60-fold fewer CFUs for the Δ*mxiD* strain, providing a wide dynamic range for the assay (Fig. 1c). In this window of early epithelial colonization, the intracellular *Shigella* population was the product of active T3SS-invasion and around four rounds of replication, with only limited IEC death elicited (Extended Data Fig. 1b,e).

Infection bottlenecks can confound genome-wide screens and generate false positive results[23,24,31]. To quantify the infection bottleneck size, we generated a random *Shigella* Tn5 mutant library with, in total, ~130,000 mutants, covering both the chromosome and pINV (Extended Data Fig. 2a). Sub-libraries containing 10,000 or 30,000 mutants were used to infect single wells of enteroids in triplicate. Because of the

limited size of the intracellular *Shigella* population, input and output libraries were enriched before sequencing. This revealed the presence of approximately 1,500–4,000 unique Tn5 mutants within the intracellular bacterial population retrieved at 6 h.p.i. (output), indicating a tight bottleneck (Fig. 1d). We next simulated output mutant library compositions for bottleneck sizes up to 6,000 bacteria by randomly selecting bacteria from the input libraries (Methods), and comparing this to the number of unique mutants recovered in the output libraries (Fig. 1e). Assuming no selection, this indicated a bottleneck size of ~2,000 bacteria at this experimental scale.

*Shigella* expresses its T3SS at 37 °C (refs. 1,8,32). This comes at a growth cost, evident from the enhanced growth of Congo Red negative (T3SS off) *Shigella* clones[33], and dissected at the molecular level in other bacteria[34–36]. Therefore, the enrichment temperature could potentially affect the mutant composition of input and output libraries due to selective growth effects. To test the global impact of enrichment temperature, we pooled the 130,000 *Shigella* Tn5 mutants and grew the resulting cultures at 30° (T3SS cascade off) or 37 °C (T3SS cascade on). Transposon insertions in a large number of genes (~1,200) led to differential fitness during enrichment at 37 °C, while approximately tenfold fewer gene disruptions (~130) affected fitness at 30 °C (Fig. 1f, Extended Data Fig. 2b–d and Supplementary Data 1). To further link bacterial fitness and T3SS virulence gene expression, we performed RNA-sequencing (RNA-seq) in a wild-type *Shigella* strain grown at 30° or 37 °C (temperature-regulated genes) and a wild-type *Shigella* strain versus a Δ*virF* mutant (lacking the master regulator of the T3SS cascade), grown at 37 °C (VirF-regulated genes) (Fig. 1g, Extended Data Fig. 2e–g and Supplementary Data 2). By combining RNA-seq (gene expression, wild-type, 37° versus 30 °C) and TraDIS (fitness, enrichment at 37° versus 30 °C) data, we quantified the genome-wide cost of virulence gene expression in *Shigella* (Fig. 1h and Extended Data Fig. 2h). Among the T3SS-associated mutants, two—namely those with disrupted *virF* and *virB*—showed significantly better growth at 37 °C. This is particularly relevant since *virF* and *virB* are master regulators of the T3SS cascade and, at 37 °C, activate the structural T3SS apparatus genes and those encoding secreted effectors[32]. By contrast, little or no cost was associated with individual T3SS structure or effector genes (Fig. 1h). Other mutants with increased fitness at 37 °C exhibited disruption of *virF* transcriptional activators (for example, Δ*fis* and Δ*cpxR* mutants)[37–39] (Fig. 1h and Extended Data Fig. 2d–h).

For validation, we developed an internally controlled competition assay, encompassing defined consortia of genetically barcoded wild-type and mutant *Shigella* strains and quantification by quantitative PCR (qPCR; Extended Data Fig. 2i,j) (informed by refs. 26,40). Here, we mixed 1:1:1:1:1:1:1 consortia, containing seven strains: three wild-type (tagA, tagB, tagC), two Δ*mxiD* (tagD, tagE) and two Δ*virF* (tagF, tagG) or two Δ*virB* (tagF, tagG) strains. All strains were represented equally in the input consortia (Fig. 1i,j). We then grew these overnight at either 30° or 37 °C. Again the Δ*virF* and Δ*virB* mutants were markedly overrepresented at 37 °C compared to wild-type and Δ*mxiD* strains, whereas no difference in relative strain abundance was observed at 30 °C (Fig. 1i,j). Altogether, we successfully mapped and quantified the cost of temperature-induced T3SS gene expression in *Shigella*, expanding previous gene essentiality studies[41]. Moreover, performing the bacterial input and output enrichment at 30 °C seems critical to avoid biased results from TraDIS experiments in *Shigella*-infected enteroids, or other infection models.

### *Shigella* genome-wide map of enteroid colonization genes

To define the *Shigella* factors required to colonize human epithelium, we scaled the enteroid infection assay to a genome-wide screen using Tn5 mutant libraries. We optimized the infection assay (Fig. 2a) based on pilot experiments (Fig. 1) to balance cost, experimental throughput and ease of analysis. Given the infection bottleneck of ~2,000 bacteria with 1,000–2,000 unique insertion sites recovered after infection

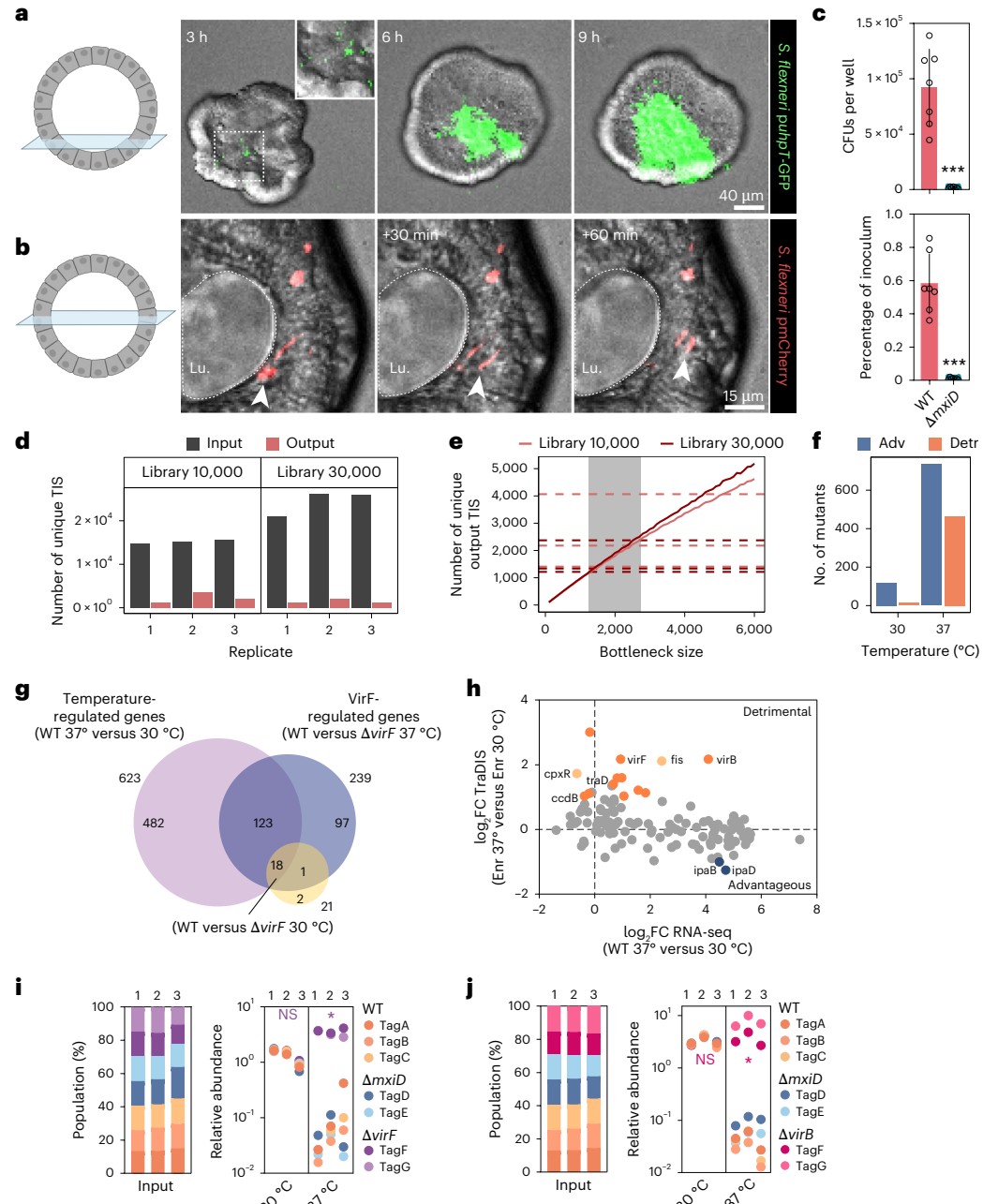

**Fig. 1 | Scalable human intestinal epithelium *Shigella* infection model.**
**a,b**, Representative time-lapse series of a BO enteroid infected with wild-type
*Shigella* harboring cytosolic reporter p*uhpT*-GFP (**a**) or constitutive reporter
p-mCherry (**b**) at MOI 40. See also Supplementary Videos 1 and 2. Cartoon
depicts the imaging plane. Experiment repeated twice with imaging of at least ten
enteroids per experiment. Lu., lumen. **c**, *Shigella* CFU counts (top) and percentage
of inoculum (bottom) upon coinfection of BO enteroids with wild-type (WT) and
Δ*mxiD* (noninvasive) strains for 6 h at MOI 40; *n* = 7 biological replicates pooled
from three independent experiments. Data shown as mean ± s.d. Significance
determined by two-sided Mann–Whitney *U* test; ****P* < 0.001. **d**, BO enteroids
were infected with a *Shigella* Tn5 random mutant library containing ~10,000
or ~30,000 mutants. Shown is the number of unique TIS mapped in input (dark
gray) or intracellular population output libraries (red). **e**, Simulations of the
number of unique TIS in output samples for bottleneck sizes up to 6,000 bacteria
(solid lines) and the measured number of unique TIS (dashed lines). Gray area
indicates the plausible range of bottleneck sizes based on the simulations.
**f**, *Shigella* Tn5 random mutant library 1 (~130,000 mutants) was grown overnight
at 30 °C or 37 °C and the mutant abundance compared by TraDIS with respect to
the subculture (Methods). Shown is the number of significant advantageous (Adv)
(blue; gene presence favors growth) or detrimental (Detr) (orange; gene presence
supresses growth) genes; log$_2$FC ≥ 1, FDR ≤ 0.01. **g**, Venn diagram of *Shigella*

differentially expressed genes in the following comparisons: wild-type, 37 °C
versus 30 °C (temperature-regulated genes), wild-type versus Δ*virF* 37 °C (VirF-
regulated genes) and wild-type versus Δ*virF* 30 °C; log$_2$FC ≥ 1, FDR ≤ 0.01.
**h**, Graph showing relative mutant fitness for pINV-located genes (log$_2$FC as
noted in Extended Data Fig. 2d; Enr 37 °C versus Enr 30 °C) versus the respective
expression changes (RNA-seq; log$_2$FC between wild-type *Shigella* 37 °C versus
30 °C; Extended Data Fig. 2e). Significant pINV advantageous (dark blue; gene
presence favors growth) or detrimental (dark orange; gene presence supresses
growth) genes are shown. Two selected chromosomal *virF* regulator genes,
*cpxR* and *fis*, also indicated in light orange; log$_2$FC ≥ 1; FDR ≤ 0.01. **i**, Barcoded
competition assay with a consortium comprising three *Shigella* wild-type
(tagA, tagB, tagC), two Δ*mxiD* (tagD, tagE) and two Δ*virF* (tagF, tagG) strains
grown overnight at 30 °C or 37 °C. Left, percentage of each strain in the input
populations. Right, quantification of relative strain abundances in the *Shigella*
barcoded consortia cultures grown at 30° or 37 °C. Relative abundances
normalized to the input inoculum. Data for three independently generated
consortia. NS, not significant. **j**, As in **i**, but using a consortium comprising three
*Shigella* wild-type (tagA, tagB, tagC), two Δ*mxiD* (tagD, tagE) and two Δ*virB* (tagF,
tagG) strains. Significance in **i** and **j** determined by two-sided paired *t*-test between
normalized output and normalized input abundances (Methods). **P* < 0.05. Panels
**a** and **b** partially created using BioRender.com.

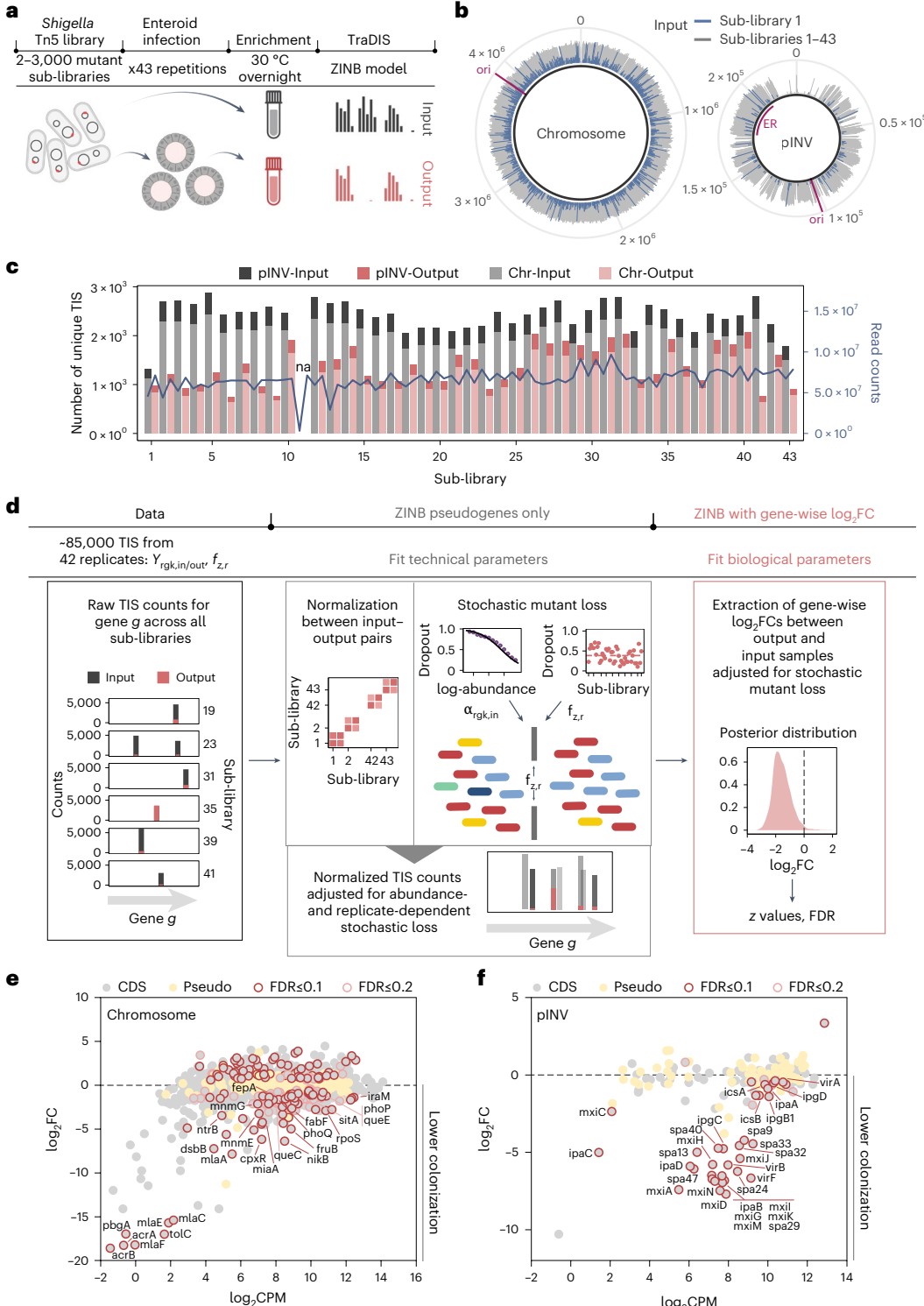

**Fig. 2 | Genome-wide map of the *Shigella* geneset required to invade the human enteroid infection model. a**, Schematic of the optimized protocol used for the *Shigella* TraDIS screen in human BO enteroids. **b**, *Shigella* M90T chromosome and pINV maps showing the distribution of TIS across the 43 input sub-libraries (gray) versus input sub-library 1 only (blue); ori, origin of replication for the chromosome and pINV; ER, entry region for the pINV. **c**, Number of unique TIS and total number of sequencing read counts across all input and output sub-libraries. Sub-library 11 was removed in the subsequent analysis due to the low number of sequencing read counts in the input sample. **d**, Schematic of the ZINB model developed to map *Shigella* genome-wide colonization factors in the presence of infection bottlenecks. Workflow shows the TraDIS raw dataset used for the sequential fitting of ZINB model parameters. The ZINB model was

first applied to TIS in pseudogenes only to fit technical parameters (pairwise normalization factors and the dependence of stochastic mutant loss on input abundance $\alpha_{rgk,in}$ and the fraction of zeros in the output sample $f_{z,r}$). These fitted parameters were then used in a second ZINB model applied to all ~85,000 TIS to extract gene-wise $\log_2$FC values and determine significance. **e,f**, MA plots showing chromosome-located (**e**) or pINV-located (**f**) *Shigella* gene mutant relative abundances upon infection of enteroids, as quantified by TraDIS. Shown is the $\log_2$FC in the output library on the *y* axis and the $\log_2$CPM (average $\log_2$-transformed counts per million) on the *x* axis as detailed in **d**; each dot represents a gene. CDS, coding sequence; Pseudo, yellow, pseudogene. Panel **a** partially created using BioRender.com.

(Fig. 1d,e), we expected sub-libraries with 2,000–3,000 unique Tn5 mutants to limit stochastic loss to about 50%. Hence, approximately 40 sub-libraries would result in coverage of at least 80% of the genome (Extended Data Fig. 3a,b and Methods) (43 sub-libraries opted for). Each sub-library was used to infect a single batch of ~800 enteroids (Fig. 2a and Extended Data Fig. 3c). At 6 h.p.i. we recovered 40,000–200,000 intracellular bacteria per infection (Extended Data Fig. 3d). We then enriched intracellular (output) and inoculum (input) bacteria for 14 h at 30 °C to minimize selection bias. As expected, the unique insertion sites of any single sub-library did not cover the genome evenly (Fig. 2b, blue), but a combination of all input sub-libraries provided complete coverage of the chromosome and pINV (Fig. 2b, gray), with sequencing reads distributed evenly across 42 of the 43 samples (Fig. 2c). This approach allowed screening of ~112,700 mutant clones (Extended Data Fig. 3c), and quantification of ~85,000 unique *Shigella* Tn5 mutants for colonization defects (by TraDIS), covering 86% of annotated genes with at least one insertion site (Extended Data Fig. 3e).

To enable robust statistical analysis, we devised a Bayesian model tailored to our experimental design (Fig. 2d, Extended Data Fig. 3f, Supplementary Note 1 and Methods). Each sub-library has a unique mutant composition, yielding a pairwise structure (Extended Data Fig. 3g) that requires normalization. We estimated gene-wise $\log_2$-transformed fold change ($\log_2$FC) across all insertion sites in a gene, assuming that all Tn5 mutants have similar effects on colonization. To handle stochastic mutant loss, we used a ZINB distribution combining the negative binomial model for count data with a zero-inflation component to capture excess zeros (Extended Data Fig. 3h). Observed zero rates (25–75%, Extended Data Fig. 3i), matched expectations from bottleneck size estimates, and increased with lower mutant abundance (Extended Data Fig. 3j). We followed a two-step process to extract global gene fitness measurements (Fig. 2d). Initially, pairwise normalization factors and technical parameters, such as the abundance-dependence of mutant loss (Extended Data Fig. 3f), were derived by fitting a ZINB model to 769 pseudogenes assumed to be neutral. Subsequently, these parameters informed a second ZINB model applied to all genes to extract gene-wise $\log_2$FCs, representing the differential abundance in the output samples adjusted for stochastic loss (Extended Data Fig. 3j). We calculated $z$ values from the $\log_2$FC posterior distributions (Fig. 2d) and assigned a false discovery rate (FDR) to correct for multiple hypothesis testing[42] (Extended Data Fig. 3k,l, Supplementary Note 1 and Methods). This allowed us to systematically evaluate the effect of gene disruption on enteroid colonization genome-wide, accounting for infection bottlenecks and maintaining a stringent FDR.

Applying this framework, we found 105 chromosomal genes to be advantageous for *Shigella* enteroid colonization (that is, mutants less invasive; 62 genes with FDR < 0.1 and 43 additional genes with FDR < 0.2) (Fig. 2e and Supplementary Data 3 and 4), a large fraction of which not previously linked to *Shigella* epithelial colonization (>70; Supplementary Data 4). Among the advantageous genes previously unreported were those encoding the stress regulator RpoS and anti-adapter protein IraM, the polyribonucleotide nucleotidyltransferase PNPase, several

transporters (NikB, FruB, NagE, ArgO, YfeH, BrnQ, YhjX, TsgA and CusA), the beta-ketoacyl-ACP synthase II FabF, tRNA-modification enzymes (MnmE and MnmG) and sulfur metabolism-associated genes (*yftE*, *cysI*, *cysQ* and *yeeE*). Seven mutants belonging to transmembrane transporter complexes (Δ*mlaC*, Δ*mlaE*, Δ*mlaF*, Δ*pbgA*, Δ*acrA*, Δ*acrB* and Δ*tolC*), were essentially lost from the output (intracellular bacteria) population. Follow-up barcoded consortium infections showed that the Δ*acrA*, Δ*acrB*, Δ*acrAB* and Δ*tolC* mutants were attenuated for enteroid colonization and also sensitive to several infection assay conditions, extending previous observations[43] (Extended Data Fig. 4a–g).

Moreover, 61 chromosomal genes decreased the colonization ability of *Shigella* (that is, mutants enriched in intracellular population; 49 with FDR < 0.1; 12 with FDR < 0.2; Fig. 2e). Among these, we identified LPS and O-antigen biosynthesis genes (*waaO*, *rfbD*, *galU*, *waaJ*, *rfbB*, *waaD*, *rfaL*, *rfbA* and *rfbC*) (Supplementary Data 4). These mutants probably harbor a shorter LPS, favoring T3SS docking to the IEC membrane[44–46], although LPS-affecting mutants may be attenuated in vivo owing to poor cell envelope protection[44–47].

On pINV, 38 genes scored as required or advantageous for enteroid colonization (34 with FDR < 0.1; 4 with FDR < 0.2) (Fig. 2f). Among them were the *virF* and *virB* transcriptional regulator genes, most genes encoding T3SS structural components (including *mxiD*, deleted in our noninvasive control strain in Fig. 1) and *icsA*, involved in actin-based intracellular spread. This comprehensive coverage of the core T3SS machinery demonstrates the high level of saturation attained in the screen. In conclusion, we successfully combined TraDIS with high-throughput enteroid infections to generate, to our knowledge, the first genome-wide map of *Shigella* genes required to colonize nontransformed human gut epithelium.

### *Shigella* T3SS effector requirements across epithelial contexts

The TraDIS screen revealed *Shigella* genes required for enteroid colonization from the basal side (Fig. 2e,f). For T3SS-related components, in addition to the IpaBCD translocon, we found a set of secreted effectors (*ipgB1*, *ipgD*, *icsB*, *ipaA* and *virA*) to promote colonization (Figs. 2f and 3a). Previous studies have explored the dependence of *Shigella* on T3SS effectors for epithelial cell colonization, typically using tumor-derived cell lines and sometimes with contrasting results (reviewed in ref. 9). Moreover, *Shigella* invasion seems less efficient from the apical than from the basal side of the epithelium[14,15,21]. In follow-up experiments, we found S*higella* intracellularly in infected human 3D enteroids and colonoids showing either BO or AO polarity (Fig. 3b–e). A similar number of wild-type *Shigella* could be recovered from BO enteroids and BO colonoids when coinfected with a mix of wild-type and Δ*mxiD* strains (Fig. 3f–h). *Shigella* colonization of AO enteroids was around fivefold less efficient, and in AO colonoids ~15-fold less efficient than in the corresponding BO counterparts (Fig. 3i). In all cases, a consistent and large (≥56-fold) difference in CFUs was evident between the wild-type and the Δ*mxiD* noninvasive strain (Fig. 3f–i).

To substantiate and extend the TraDIS results (Figs. 2 and 3a), we quantified differences in IEC colonization efficiency of wild-type

**Fig. 3 | T3SS effector contributions to *Shigella* IEC colonization across different cell geometries and intestinal segments. a**, Heat map showing $\log_2$FC for T3SS effector gene mutants, as informed by the TraDIS screen in Fig. 2f; **FDR < 0.1; *FDR < 0.2. **b–e**, Representative confocal fluorescent microscopy images of a BO enteroid (**b**), AO enteroid (**c**), BO colonoid (**d**) and AO colonoid (**e**) infected with wild-type *Shigella* harboring cytosolic reporter p*uhpT*-GFP at MOI 40. Experiments were repeated at least twice with imaging of at least ten enteroids or colonoids per experiment. **f–i**, *Shigella* CFU counts (left) and percentage of inoculum (right) upon coinfection with wild-type and Δ*mxiD* (noninvasive) strains for 6 h at MOI 40 of BO enteroids (**f**, same plots as in **c**), AO enteroids (**g**), BO colonoids (**h**) and AO colonoids (**i**); $n = 7$ biological replicates pooled from two to three independent experiments. Data shown as mean ± s.d. Significance determined by two-sided Mann–Whitney $U$-test; ***$P < 0.001$.

**j–m**, BO and AO enteroids (**j**) and BO and AO colonoids (**l**) were infected with a barcoded consortium comprising two wild-type (tagA, tagB), two Δ*mxiD* (tagC, tagD) and two mutant (tagE, tagF) strains at MOI 40 for 6 h. Shown is the quantification of relative tag abundance, normalized against the corresponding input (Extended Data Fig. 5a,b). Data represent two independently generated consortia for each infection. Significance determined by two-sided paired $t$-test for each specific mutant between normalized output and normalized input abundances (Methods). *$P < 0.05$, **$P < 0.01$, ***$P < 0.001$. **k–m**, Bargraph showing the colonization index for each mutant in BO (dark gray) and AO (light gray) enteroid infections (**k**, derived from data in **j** and calculated as 1 − (WT/mutant); $n$ as in **j**) and BO (dark gray) and AO (light gray) colonoid infections (**m**, derived from data in **l**; $n$ as in **l**). Data shown as mean ± s.d. Wild-type shown as 0.

*Shigella* and clean T3SS effector mutants by barcoded consortium infections in BO enteroids (same condition as the TraDIS screen), as well as AO enteroids, BO colonoids and AO colonoids. For each infection, an equally mixed (1:1:1:1:1:1) inoculum of six strains, two wild-type (tagA, tagB), two Δ*mxiD* (tagC, tagD) and two mutant strains for a given effector (tagE, tagF) was prepared. This setup thus used two biological

replicates for each T3SS genotype, allowing powerful internally controlled comparisons. We first infected BO and AO enteroids. All strains were represented equally in the input inocula (Extended Data Fig. 5a). In the intracellular population, the two noninvasive Δ*mxiD* control strains were approximately 100-fold to 1,000-fold less abundant than the two wild-type strains (Fig. 3j). To visualize colonization differences

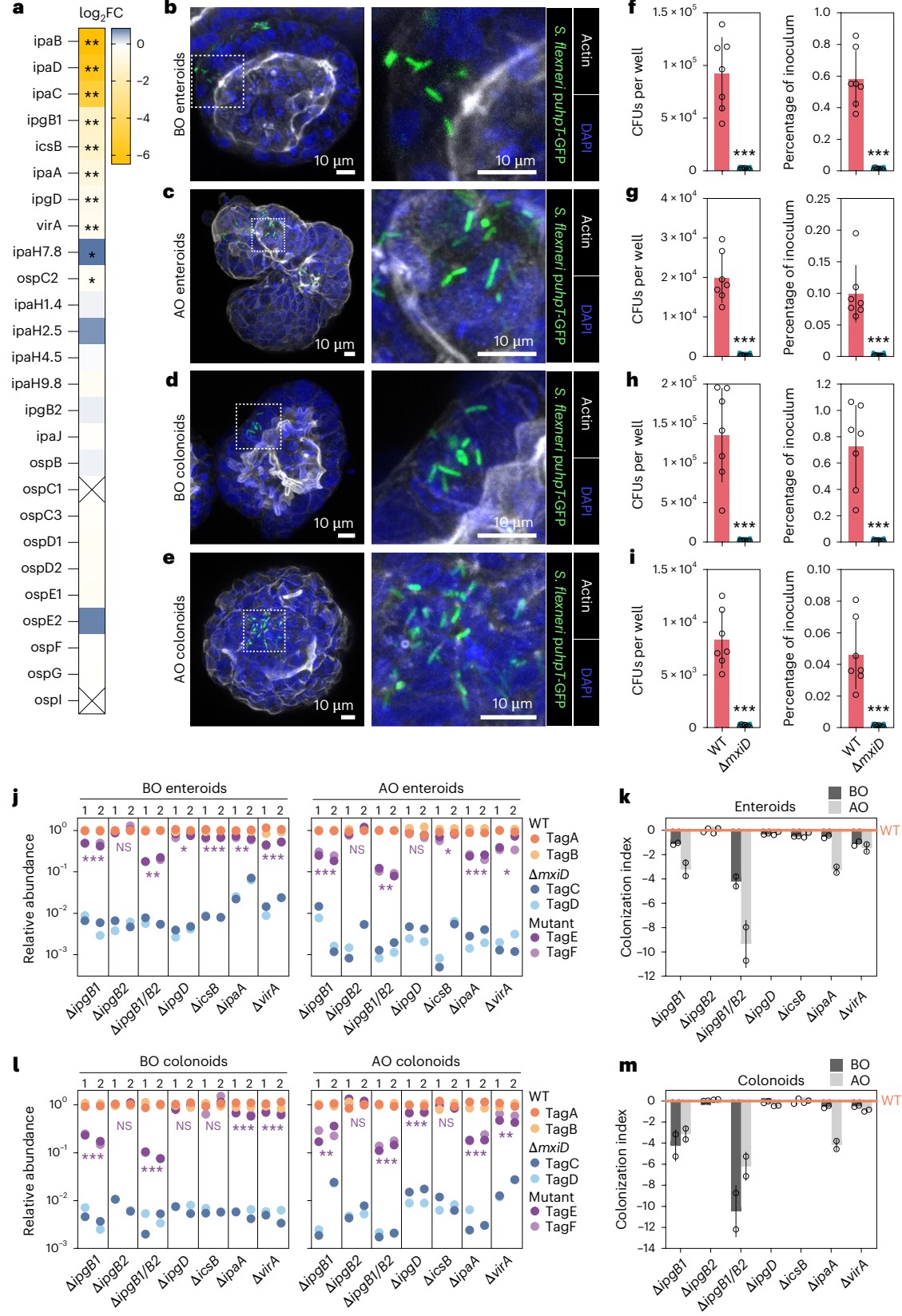

between strains, we calculated a colonization index for each mutant (1 – (mean relative abundance$_{WT}$/mean relative abundance$_{mutant}$)) in relation to wild-type *Shigella* (value 0). Deletion of *ipgD* and *icsB* resulted in a minimal but significant colonization reduction in BO enteroids. In AO enteroids, deletion of *ipgD* did not cause significant reduction of colonization, whereas *icsB* deletion again showed a minor contribution (Fig. 3j,k). By contrast, deletion of *ipgB1* attenuated colonization (around twofold in BO and around fourfold in AO enteroids), and the combined deletion of IpgB1 and IpgB2 (Δ*ipgB1/B2*) even more so (around fourfold in BO and greater than tenfold in AO enteroids). Deletion of IpaA (Δ*ipaA*) had a minor impact in BO enteroids (mutant ~30% less invasive than wt), but caused an attenuation in AO enteroids of about fivefold. Moreover, deletion of VirA caused an attenuation in both BO and AO enteroids of approximately twofold. We next used the same consortia to infect BO and AO colonoids (Fig. 3l,m and Extended Data Fig. 5b). Here, deletion of *ipgB1* and *ipgB1/B2* had an even bigger effect on *Shigella* colonization. A negligible contribution could again be observed for IpgD and IcsB and a modestly stronger one for VirA (Fig. 3l,m). Finally, deletion of IpaA caused markedly attenuated colonization, particularly in AO colonoids, similar to the enteroid infections (compare Fig. 3l,m with Fig. 3j,k). This phenotype was also validated in additional enteroid and colonoid lines from independent donors (Extended Data Fig. 5c–f).

We conclude that the TraDIS screen captured even minor contributions of *Shigella* genetic elements to the colonization process (compare Figs. 2f and 3a to Fig. 3j; BO enteroids). Moreover, barcoded infections reveal that a minimal set of T3SS effectors required for early *Shigella* colonization of a nontransformed human intestinal epithelium includes four main effectors: IpgB1 and IpgB2 (Rho GTPase GEFs; see refs. 48–50), IpaA (a vinculin-binding and actin-depolymerizing protein; see refs. 51,52) and VirA (a Rab GAP; see refs. 53,54). We found no obvious difference in the overall effector requirement for colonization of jejunum- or colon-derived IECs. However, our results identify a context-dependent requirement for IpaA specifically during apical epithelial invasion.

## MnmE/G tRNA-modification enzymes exert global virulence control

We next scrutinized the TraDIS results with a focus on chromosomal genes (Fig. 2e and Supplementary Data 3 and 4). Clean barcoded *Shigella* mutants for select genes were constructed and BO enteroid infections with mixed consortia conducted as above, which replicated weak hits (for example *fabF*, *yfeH*), and nonsignificant findings from the TraDIS (for example, for *cysK*, *fliM*, *flgA* and *fliH*) (Extended Data Fig. 6a–c).

Most interestingly, the TraDIS results suggested the tRNA-modification enzymes MnmE and MnmG to promote *Shigella* enteroid colonization (Fig. 2e and Supplementary Data 3 and 4). The MnmEG complex modifies position 5 of the wobble uridine (U34) in the anticodon of specific tRNAs, namely tRNA$^{Arg}$-mnm5UCU, tRNA$^{Gly}$-mnm5UCC, tRNA$^{Leu}$-cmnm5UmAA, tRNA$^{Gln}$-cmnm5s2UUG, tRNA$^{Glu}$-mnm5s2UUC and tRNA$^{Lys}$-mnm5s2UUU[55,56] (Fig. 4a). The related enzymes MnmA and MnmC drive further U34 modifications[55,56]. Tn5 insertions in *mnmA* were rare in the input, whereas *mnmC* insertions did not affect *Shigella* colonization (Supplementary Data 3). By contrast, the Δ*mnmE* and Δ*mnmG* mutants showed a slight growth defect (Extended Data Fig. 6d) and a striking six- to tenfold lower colonization capacity than wild-type strains in barcoded consortium infections, and in CFU plating assays (Fig. 4b and Extended Data Fig. 6e).

Suspecting that MnmE and MnmG act through translation effects, we conducted proteome profiling. For the 2,153 identified proteins in the Δ*mnmE* mutant, 154 proteins were significantly downregulated, and 71 significantly upregulated in comparison to wild-type strains (Fig. 4c and Supplementary Data 5; log$_2$FC ≥ 0.5; adjusted *P* value ≤ 0.01). Essentially identical results were obtained for the Δ*mnmG* proteome (158 downregulated; 72 upregulated; Fig. 4d, Extended Data Fig. 6f,g and Supplementary Data 5). Among the downregulated proteins, 25 were virulence proteins encoded on pINV, including the VirB transcriptional regulator, T3SS structural components (for example, IpaB, IpaC, MxiA, MxiC, MxiH, MxiN, SpaM, Spa33 and Spa47), early effectors (for example, IpaA and IpgB2) and late effectors (for example, OspC2, OspC3, OspF and PhoN2) (Fig. 4c, d). This offers a compelling explanation for the reduced colonization capacity of Δ*mnmE* and Δ*mnmG* strains.

The MnmE/G U34-dependent tRNA modifications xm5U ensure translation accuracy and fidelity. They stabilize base pairing between codons ending in A or G (NNA and NNG) with the corresponding anticodons, while hampering base pairing with codons ending in C and U[57]. This led us to ask whether codon usage correlates with *Shigella* virulence protein expression and its dependence on *mnmE* and *mnmG*. Analysis across the *Shigella* genome revealed that pINV genes showed higher frequency for AGA-Arg, AGG-Arg, GGA-Gly and AUA-Ile codons when compared to chromosomal genes (Extended Data Fig. 6h). Notably, most of the downregulated virulence proteins in the tRNA modification mutants (21 out of 25) showed an even higher AGA-Arg codon usage ratio than the average ratio (0.19) of all pINV genes. This is particularly interesting since the arginine AGA codon-reading anticodon is modified by MnmE/G and AGA is among the rarest codons in *Escherichia coli*[58]. Indeed, this difference was even more evident when comparing the AGA codon usage ratio for virulence proteins with the average ratio (0.04) for

**Fig. 4 | Global regulation of pINV-encoded virulence proteins via tRNA modification and differential codon abundance. a**, Schematic of MnmE/G-dependent tRNA modifications. **b**, BO enteroids were infected with barcoded consortia comprising two wild-type (tagA, tagB), two Δ*mxiD* (tagC, tagD) and two mutant (tagE, tagF) strains at MOI 40 for 6 h. Left, quantification of relative tag abundance, normalized against the corresponding input (Extended Data Fig. 6d). Data for three independently generated consortia for each infection. Significance determined by two-sided paired *t*-test between normalized output and normalized input abundances. **P < 0.01. Right, colonization index for the Δ*mnmE* and Δ*mnmG* mutants (derived from data in left panel). Data shown as mean ± s.d. **c,d**, Volcano plots showing differentially abundant proteins in *Shigella* Δ*mnmE* versus wild-type (**c**) or Δ*mnmG* versus wild-type (**d**) strains. Each dot represents a protein. Differentially abundant proteins encoded on pINV shown in pink, on chromosome in black and all nonsignificant differentially abundant proteins in gray. Significance determined by two-sided limma analysis with Benjamini–Hochberg multiple comparison correction; log$_2$FC ≥ 0.5; adjusted *P* value ≤ 0.01. **e,f**, Plots showing relative protein levels (log$_2$FC as in **c**, Δ*mnmE* versus wild-type) versus AGA codon usage ratios for Arg (**e**) and GGA codon usage ratios for Gly (**f**) per protein. Differentially abundant proteins encoded on pINV shown in pink, on chromosome in black, and nonsignificant differentially abundant proteins in gray; log$_2$FC ≥ 0.5; adjusted *P* value ≤ 0.01.

Dark and light green dashed lines specify mean AGA/GGA codon usage ratios for all open reading frames on pINV or chromosome, respectively. **g–i**, *ipaA* (**g**), *mnmE* (**h**) and *mnmG* (**i**) mRNA levels (2$^{-\Delta\Delta Ct}$) upon induction with 0.25 mM IPTG in the indicated strains carrying an inducible *ipaA-3xFT* plasmid, p-Empty or p-*mnmE* or p-*mnmG*, normalized to the corresponding mRNA levels in wild-type *Shigella*. **j**, Relative IpaA-3xFT protein levels (0.25 mM IPTG) in wild-type *Shigella*, Δ*mnmE* or Δ*mnmG* strains carrying plasmids as in **g–i**. Quantification by western blot of serially diluted samples with *wt*/*ipaA-3xFT* protein levels set as 1. Data shown as mean ± s.d. of three independent experiments (Extended Data Fig. 7a,b). **k,l**, BO enteroids were infected with barcoded consortia comprising the indicated strains at MOI 40 for 6 h. Shown is the quantification of relative tag abundance, normalized on the corresponding input (Extended Data Fig. 7c,d). Data for two independently generated consortia per infection. Significance determined by two-sided paired *t*-test between normalized output and normalized input abundances. Only consortia including the two mutant strains carrying the p-Empty or p-*mnmE* (**k**) p-*mnmG* (**l**) plasmids were used in the analysis. *P < 0.05, **P < 0.01, ***P < 0.001. **m,n**, *mnmE* (**m**) and *mnmG* (**n**) mRNA expression (2$^{-\Delta\Delta Ct}$) in the indicated strains, normalized to *mnmE* (**m**) and *mnmG* (**n**) expression in wild-type *Shigella*. For **g,h,i,m** and **n**, *n* = 3 biological replicates. Data shown as mean ± s.d. Panel **a** partially created using BioRender.com.

genes on the *Shigella* chromosome (Fig. 4e and Extended Data Fig. 6i). Similar observations were true for the GGA-Gly codon usage ratio (Fig. 4f and Extended Data Fig. 6j). Among other codons modified by the MnmE/G complex, we also observed a mild enrichment for UUA-Leu

and CAA-Gln codon usage ratio for downregulated virulence proteins, but no evident enrichment for GAA-Glu and AAA-Lys (Extended Data Fig. 6k–n). Moreover, AGG-Arg and GGG-Gly codons, which can potentially pair with MnmE/MnmG-modified U34 of tRNA$^{Arg}$-mnm5UCU and

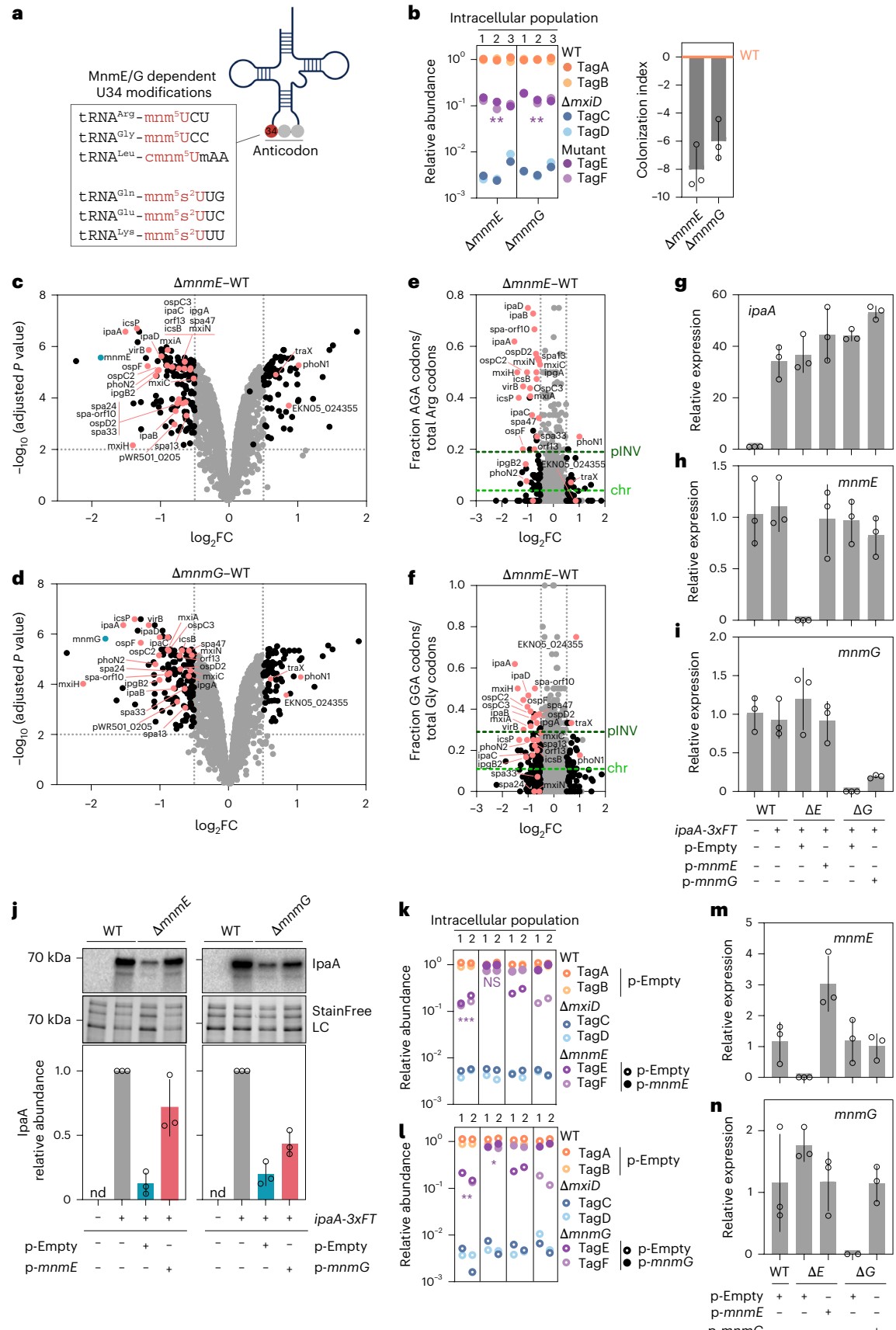

tRNA$^{Gly}$-mnm$^5$UCC, were also slightly enriched among the downregulated virulence proteins (Extended Data Fig. 6o,p).

To formally test the link between MnmE/G U34 tRNA modifications and *Shigella* virulence protein expression, we selected IpaA as a prototypical T3SS protein with a high AGA and GGA codon usage (Fig. 4e,f). We assessed *ipaA*/IpaA mRNA and protein production following inducible induction with isopropyl β-ᴅ-1-thiogalactopyranoside (IPTG), finding comparable mRNA levels across all strains tested (Fig. 4g). However, IpaA protein levels were decreased strongly in the Δ*mnmE* and Δ*mnmG* mutants (Fig. 4j and Extended Data Fig. 7a,b). This indicates a direct MnmE/G effect on translation of AGA- and GGA-enriched transcripts. Complementation of *mnmE*/*G* restored IpaA protein to wild-type levels in the Δ*mnmE* mutant and partially so in the Δ*mnmG* mutant (Fig. 4j,h,i and Extended Data Fig. 7a,b). Complementation also completely restored the enteroid colonization defect of the Δ*mnmE* mutant, and to a high extent for the Δ*mnmG* mutant (Fig. 4k–n and Extended Data Fig. 7c,d; input). Hence, the combination of enteroid infections with a *Shigella* genome-wide TraDIS screen allowed us to identify a global mode of *Shigella* virulence control governed by selective codon usage and the MnmE/MnmG tRNA-modification enzymes.

## Discussion

Historically, infection experiments aspiring to mimic host physiology have been possible using in vivo models or short-lived explants. Despite much recent progress[59,60], no small animal model exists that fully recapitulates human shigellosis[3]. Organotypic tissue culture has opened new possibilities to explore microorganisms–host interactions while capturing central aspects of host cell and tissue physiology[13–20,61–63]. Here, we combined TraDIS with high-throughput infection assays in 3D enteroid/colonoid suspension cultures[16,22] to map the comprehensive geneset *Shigella* uses to colonize human intestinal epithelium. Among limitations, it should be noted that the enteroids/colonoids still lack innervation, blood-derived cells and vasculature, and that the short infection window of the screen limits the capture of phenotypes linked to immune evasion and cell death suppression. Nevertheless, our effort provides a key proof-of-principle for functional genomics approaches in organotypic infection models that future studies can build on for diverse pathogens and time-scales.

Common concerns when using TraDIS and similar screening techniques in an infection setting are: (1) the impact of enrichment of bacterial input and output populations and (2) the consequences of infection bottlenecks, which result in stochastically reduced genetic diversity[24,64]. We addressed these concerns to ensure the success of the screen. Combining RNA-seq (gene expression) and TraDIS (fitness), we quantified the condition-specific cost of virulence gene expression in *Shigella*. Besides providing a framework for measuring the effect of virulence gene expression on bacterial fitness in a genome-wide fashion, this analysis provided a key premise for the main TraDIS screen: performing enrichment growth step(s) at 30 °C instead of 37 °C leads to minimal bias in defining the *Shigella* geneset requirement for epithelial colonization.

Infection bottlenecks limit forward genetic screens in complex models[24,65–69], including a previous screen using an infant rabbit shigellosis model[64]. To overcome this obstacle, we assessed bottlenecks experimentally and using computational simulations. Performing about 40 infections with *Shigella* Tn5 mutant sub-libraries with 2,000–3,000 mutants each achieved a balance between reducing bottleneck effects and experimental feasibility. Previous ZINB models to describe stochastic loss in single-cell RNA-seq[70,71] and Tn-seq experiments[72] fit a gene-dependent zero-inflation parameter, conflating stochastic loss with biological zeros. Using nonfunctional pseudogenes, we showed that stochastic loss depends on the input abundance of each mutant and varies between replicates. Our Bayesian model incorporates these gene-independent effects, enabling integration across experiments and identification of genuine phenotypes. The combination of experimental and computational optimization hence defined a *Shigella*

epithelial infectome of 105 chromosomal and 38 pINV genes, a large fraction of which not previously linked to virulence.

Key findings from the TraDIS screen were validated and expanded upon by our *Shigella* barcoded consortium infection technology[26,27], applied to BO and AO enteroids and colonoids[16,22]. This highlighted, for example, an impact of four main T3SS effectors: the actin ruffle-inducers IpgB1 and IpgB2 (refs. 48–50), the actin-depolymerization protein IpaA[51,52] and the Rab inhibitor VirA[53,54] for early IEC colonization across segments and geometries. Moreover, we observed a context-dependent requirement for IpaA specifically for *Shigella* apical invasion of enterocytes and colonocytes. Follow-up studies may decipher how these T3SS activities are coordinated, and how the IpaA apical invasion phenotype relates to the vinculin-binding property of this effector[51,52].

Among the >100 chromosomal *Shigella* genes required for enteroid colonization, we focused on *mnmE* and *mnmG*, two tRNA-modification enzymes modifying U34 of various tRNA species[55,56]. Previous reports have linked virulence phenotypes to MnmE/G tRNA modifications in other bacteria[73,74]. In addition, MnmA contributed to *Mycobacterium tuberculosis* intracellular growth in macrophages[75], while MiaA favored extraintestinal pathogenic *E. coli* (Ex-PEC) in vivo infections[76]. In *Shigella*, Tgt and MiaA have been shown to impact VirF translation[77,78], and TruB slightly affected IpaB expression[79]. However, global effects of tRNA modifications on virulence gene expression have remained elusive. We show that pINV-encoded virulence proteins are globally downregulated in MnmE/G mutants. This reduction is associated with an increased fraction of AGA and GGA codons in the corresponding transcripts. Both AGA and GGA codons are translated by tRNAs carrying MnmE/G-dependent modifications[80]. This suggests direct MnmE/G-dependent global translation control of AGA and GGA codon-enriched transcripts encoded on pINV. Such stringent virulence expression control is crucial to balance virulence potential with the fitness cost of expressing these genes. The nucleoid associated protein H-NS silences transcription of horizontally acquired AT-rich sequences in bacterial pathogens[37,81,82], and in *Salmonella* H-NS deletion results in severe fitness defects owing to aberrant T3SS expression[83]. In *Yersinia*, T3SS expression leads to growth arrest, with control mediated by virulence plasmid copy number regulation[84,85]. As shown here, expression of the *Shigella* virulence machinery causes growth defects at the inducing temperature in vitro, again highlighting the need for fine-tuned regulation. The MnmE/G-dependent regulation uncovered in *Shigella* represents a third globally acting mechanism for virulence gene expression control in an enterobacterium, made possible by differential codon usage between chromosomal genes and pINV-located virulence genes. Whether, and how, this regulatory circuit is modulated by environmental cues remains an intriguing topic for future studies.

Resolving molecular infection mechanisms is crucial to eradicate bacterial pathogens. Here, we derived a genome-wide map of *Shigella* genes required for colonization of a species-specific intestinal epithelial model by combining enteroid/colonoid infections, TraDIS, computational modeling, barcoded consortium infections and proteomics. This offers a generalizable framework to conduct genome-wide screens in complex infection models.

## Online content

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

[1]Department of Medical Biochemistry and Microbiology, Uppsala University, Uppsala, Sweden. [2]Helmholtz Institute for RNA-based Infection Research (HIRI), Helmholtz Centre for Infection Research (HZI), Würzburg, Germany. [3]Department of Surgical Sciences, Uppsala University, Uppsala, Sweden. [4]Department of Medical Sciences, Uppsala University, Uppsala, Sweden. [5]Department of Chemistry, Umeå University, Umeå, Sweden. [6]The Laboratory for Molecular Infection Medicine Sweden (MIMS), Umeå University, Umeå, Sweden. [7]Faculty of Medicine, University of Würzburg, Würzburg, Germany. [8]Department of Biology, University of Toronto, Mississauga, Ontario, Canada. [9]Department of Cell and Systems Biology, University of Toronto, Toronto, Ontario, Canada. [10]Science for Life Laboratory, Uppsala, Sweden. [11]Present address: Department of Biology and Biotechnologies 'Charles Darwin', Sapienza University of Rome, Rome, Italy. ✉e-mail: ml.dimartino@imbim.uu.se; laura.jenniches@helmholtz-hiri.de; lars.barquist@helmholtz-hiri.de; mikael.sellin@imbim.uu.se

## Methods

### Ethics statement

Human adult stem cell-derived enteroids/colonoids were established from jejunal tissue resected during bariatric surgery (enteroids), or morphologically normal nontumor colon tissue resected during elective colon cancer surgery (colonoids), in all cases following the subject's informed written consent. To ensure anonymity of tissue donors, samples were pseudonymized. All procedures were approved by the Swedish National Governing Body (Etikprövningsmyndigheten) under license/protocol 2010-157, 2010-157-1, 2020-05754 and 2023-01524-01.

### Bacterial strains, plasmids and general procedures

Bacterial strains are listed in Supplementary Table 1. M90T, a *Shigella flexneri* serotype 5 strain[7] was used as the wild-type strain and for all mutant construction. For construction of wild-type and ΔmxiD *Shigella* barcoded strains, TagA-G were integrated individually into an inert spot of the M90T chromosome, between genes *ybhC* and *ybhB*. A fragment containing an individual 40-bp tag and the chloramphenicol resistance cassette was amplified from *Salmonella* Typhimurium barcoded strains[40] and used for chromosomal integration by a modified Lambda Red Recombination protocol. For the construction of Δ*virF*, Δ*virB*, Δ*yfeH* and Δ*yejB* barcoded strains, tagF and tagG were transferred by P1 transduction from donor wild-type *Shigella* barcoded strains, followed by selection on LB agar containing 12.5 µg ml⁻¹ chloramphenicol. For all other *Shigella* barcoded mutants, the desired deletion was introduced by Lambda Red recombination in the relevant wild-type *Shigella* barcoded strains[86]. p-Empty is a de novo synthetized cloning vector carrying p15A ori and AmpR. In vitro synthesized DNA fragments containing *mnmE* and *mnmG* including their native promoters were cloned in the BamHI site of the p-Empty vector, obtaining p-*mnmE* and p-*mnmG*. pTac-28 is a de novo synthesized IPTG-inducible expression vector carrying a Ptac promoter and a *lacI* gene, together with a pMB1 ori and CmI resistance. An in vitro synthesized *ipaA-3xFT* DNA fragment was cloned into the BamHI site of the pTac-28 vector, resulting in the pTac-ipaA-3xFT plasmid. Bacteria were grown in Luria Bertani (LB) broth at 37 °C, unless otherwise specified. When necessary, antibiotics were supplemented at the following concentrations: chloramphenicol, 12.5 µg ml⁻¹; kanamycin, 50 µg ml⁻¹; ampicillin, 50 µg ml⁻¹. All plasmids and oligonucleotide primers used in this study are listed in Supplementary Tables 2 and 3, respectively.

### *Shigella* Transposon 5 library construction

Two *Shigella* Tn5-libraries were generated using EZ-Tn5 <KAN-2>Tnp Transposome kit (Lucigen, TSM99K2). In brief, 50 µl aliquots of *Shigella* M90T wt electrocompetent cells were electroporated with 1 µl of EZ-Tn5 <KAN-2>Tnp Transposome. Each reaction was recovered in 1 ml SOC (2% tryptone, 0.5% yeast extract, 10 mM NaCl, 2.5 mM KCl, 10 mM MgCl₂, 10 mM MgSO₄, 20 mM glucose) for 1 h at 37 °C, and plated on LB agar plates containing 50 µg ml⁻¹ kanamycin. For library 1 (used in the pilot screen and under different growth conditions), on the following day transformants were scraped off the LB plates and pooled into 23 sub-libraries of approximately 4,000–7,000 transformants for a total of approximately 130,000 mutants. For library 2 (used in the main TraDIS screen), transformants scraped off the LB plates were pooled into 43 sub-libraries of approximately 2,000–3,000 transformants for a total of ~112,700 mutants (Extended Data Fig. 3c).

### Enteroid and colonoid maintenance culture

Human jejunal enteroids (pseudonym ID, 18-9jej; 22-2jej) and human colonoids (pseudonym ID, 21-9col; 22-4col) were used in this study and handled as previously described[13,62,63]. For maintenance cultures, human enteroids/colonoids embedded into Matrigel domes (Corning, 356230) were subcultured weekly by mechanical dissociation and incubation with gentle cell dissociation reagent (StemCell, 07174). The resulting fragments were washed with Dulbecco's modified Eagle's medium (DMEM-F12; Gibco, 11039021) containing 0.25% bovine serum albumin (BSA; Gibco; 15260-037) and re-embedded into Matrigel domes at 1:4–1:10 splitting ratio. Cultures were kept at 37 °C in 5% CO₂, and OGM growth medium (StemCell, 06010) exchanged every 2–4 days.

### Enteroid and colonoid suspension cultures

Medium sized enteroids/colonoids were extracted by gentle dislodgment of the Matrigel domes and incubation in Cell Recovery Solution (Corning, 354253) for minimum 1 h on ice on a rotating table. Subsequently, enteroids/colonoids were allowed to sediment by gravity and the supernatant removed. After washing with DMEM-F12/0.25% BSA, the pellet was resuspended in OGM growth medium containing 8% cold Matrigel to maintain basolateral polarity (for BO) or without Matrigel to promote eversion (for AO) and aliquoted in ultra-low attachment 24-well tissue culture plates (Corning Costar, CLS3473-24EA). BO and AO suspension cultures were incubated at 37 °C with 5% CO₂ for 1 day or 3–4 days before infection, respectively.

### Enteroid and colonoid bulk infections

Shortly before infection, BO or AO enteroids/colonoids were transferred into 40 µm or 25 µm mini-cell strainers (Funakoshi, HT-AMS-12502, HT-AMS-14002) and washed three times with DMEM-F12/0.25% BSA. Enteroids/colonoids were resuspended in OGM and aliquoted in ultra-low attachment 24-well tissue culture plates. The indicated *Shigella* strains or Tn5-sub-libraries were grown overnight in LB broth containing appropriate antibiotics at 30 °C, diluted 1:50, and subcultured for 2 h at 37 °C without antibiotics. Strains were diluted in OGM medium (unless otherwise indicated) to achieve the desired MOI. To generate barcoded consortia, the indicated tagged strains were mixed in equal ratios. Bacteria were added to each well and spun down at 300g for 10 min. At 1 h.p.i., enteroids/colonoids were transferred into 25 µm mini-strainers, washed four times with DMEM-F12/0.25% BSA and incubated with medium containing 200 µg ml⁻¹ gentamicin (Sigma, G1914) for 2 h, unless otherwise indicated. For infections extending 3 h incubation, medium was replaced with 20 µg ml⁻¹ gentamicin medium up to 6 h.p.i. Infected enteroids/colonoids were washed six times with DMEM-F12/0.25% BSA, recovered from the strainers and lysed in 0.1% Na-deoxycholate (unless otherwise indicated) by vigorous pipetting or by homogenization with a Tissue Lyser (Qiagen). For live imaging, BO organoids were transferred to eight-chamber slides (Cellvis, C8-1.5H-N) and when indicated 1.5 µM DRAQ7 (Invitrogen, D15106) and/or 0.2% Saponin (Calbiochem, 558255-100) was supplemented to visualize dead or permeabilized cells. For CFU counting, intracellular bacterial populations were serially diluted and plated on LB plates containing appropriate antibiotics. For the TraDIS screen infections and barcoded infection assays, intracellular bacterial populations were enriched overnight at 30 °C in 2 ml LB broth (unless otherwise indicated). A diluted culture of the inoculum was enriched in parallel and used as the input reference.

### Caco-2 cell culture and infections

Caco-2 cell culture (ATCC HTB-37) and infections were performed as in Supplementary Note 2.

### Barcoded assays—tag quantification

For barcoded competition assays and barcoded infections, the indicated tagged strains were mixed in equal ratios. Genomic DNA from the 'input consortium' and the 'output consortium' (following growth under the indicated conditions, or following an infection assay) was extracted using the GenElute Bacterial Genomic DNA kit (Sigma, NA2110-1KT) or the DNeasy Blood and Tissue Kit (Qiagen, 69504). For tag quantification, qPCR was performed using Maxima SYBR green/ROX qPCR master mix (2×) (ThermoFisher Scientific, K0222) on a CFX384 Touch Real-Time PCR Detection System (Bio-Rad), using 9 ng

gDNA and tag-specific primers (Supplementary Table 3 and Extended Data Fig. 2i). Relative abundance of each strain was normalized to abundance in the inoculum. Standard curves were generated using gDNA from each tagged wild-type *Shigella* strain (Extended Data Fig. 2j). For barcoded infection assays, a colonization index was calculated as $1 - ($mean relative abundance$_{WT}/$mean relative abundance$_{mutant})$. To determine statistical significance, we applied a two-sided paired *t*-test to the relative input and output abundances of each strain normalized by the respective mean wild-type abundance across all barcoded consortium infections or competition assays.

## Live-cell microscopy

BO enteroids infected with *Shigella* harboring the p*uhpT*-GFP or the pmCherry plasmid, were imaged on a custom-built microscope, based on a Nikon Eclipse Ti2 body fitted with either ×40/0.6, or ×60/0.7, PlanApo air objectives (Nikon) and a Prime 95B 25 mm camera (Photometrics). The microscope chamber was maintained at 37 °C with 5% $CO_2$. Bright-field images were acquired using differential interference contrast, and fluorescence was imaged using the 475/34, 575/35 or 648/20 excitation channels of light engine Spectra-X (Lumencor) and emission collected through quadruple bandpass filters (Chroma, 89402 and 89403). Focus was maintained over time using the Perfect Focus System (PFS). Imaging started 2–2.5 h.p.i. and images were acquired every 5 min.

## TraDIS pilot screen in BO enteroids

Sub-libraries from the *Shigella* Tn5 library 1 were pooled to generate three replicate sub-libraries containing approximately 10,000 or 30,000 mutants each. The resulting sub-libraries were used to infect replicate wells containing ~500 BO enteroids at MOI 200 as described above. After infection, enteroids were lysed in 0.1% Na-deoxycholate by extensive pipetting. Intracellular bacterial populations were enriched overnight at 37 °C in 2 ml LB broth. A diluted culture of the inoculum was also enriched and used as the input reference; 2 ml of the bacterial culture was used to extract genomic DNA. A fraction of the pooled libraries, undergoing no further growth, was used to extract gDNA for TraDIS sequencing and used as Minimal Libraries reference; 2 µg of gDNA were used to prepare single 5′ sequencing libraries primed off the 5′ end of Tn5 for each sample. gDNA shearing, clean up, end repair, 'A' tailing, adapter ligation and PCR amplification were performed as detailed in ref. 87. Two pools of eight libraries were sequenced on a MiSeq instrument using v.3 150-cycle kits, run as 1 × 142 bp using a custom recipe with ten dark cycles as detailed in ref. 87.

## TraDIS under different growth conditions

The 23 sub-libraries from the *Shigella* Tn5 library 1 were pooled to generate a dense library containing ~130.000 mutants; 200 µl of the pooled library, undergoing no further growth, were used to extract gDNA for TraDIS sequencing and used as Minimal Library reference. Then, 13 µl from the pooled library (~$10^9$ bacteria) was inoculated into 200 ml LB broth with 50 µg ml$^{-1}$ kanamycin and cultures were incubated at 30 °C, 180 rpm, for 16 h. The following day, the overnight culture was subcultured 1:50 in 10 ml LB broth without antibiotics (~$2 × 10^8$ bacteria) at 37 °C for 2 h. Subsequently, 50 µl aliquots of the subculture (~$1.25 × 10^7$ bacteria) were grown overnight (12 h) in LB broth without antibiotics at either 30 °C or 37 °C, to mimic the growth enrichment step that bacteria would undergo following infection experiments. Each passage was done in triplicate. At each step, 1.5 ml culture fractions were collected and used for gDNA extraction. gDNA isolation was performed using the GenElute Bacterial gDNA kit; 2 µg of gDNA was used to prepare single 5′ sequencing libraries primed off the 5′ end of Tn5 for each sample, as in ref. 87. gDNA shearing, clean up, end repair, 'A' tailing, adapter ligation and PCR amplification were performed as in ref. 87. A single pool of all libraries was sequenced on a NextSeq 500 instrument with a 150-cycle Mid Output kit and 25% PhiX spike-in.

## RNA-seq and analysis

*Shigella flexneri* wild-type M90T and M90T Δ*virF* strains were grown in LB broth at 30 °C or 37 °C to an $OD_{600}$ of ~0.7, in triplicate. Total RNA was extracted with acid phenol as detailed in Supplementary Note 3. Total RNA was DnaseI-treated for 30 min at 37 °C and ribosomal RNA were depleted. An oligonucleotide adapter was ligated to the 3′ end of the RNA molecules. First-strand cDNA synthesis was performed using M-MLV reverse transcriptase and the 3′ adapter as primer. A 5′ Illumina TruSeq sequencing adapter was ligated to the 3′ end of the antisense cDNA. The resulting cDNA was PCR-amplified to about 10–20 ng µl$^{-1}$ using a high-fidelity DNA polymerase. The cDNA was purified using the Agencourt AMPure XP kit (Beckman Coulter Genomics). For Illumina NextSeq sequencing, the samples were pooled in approximately equimolar amounts. The cDNA pool was size fractionated in the size range of 250–600 bp using a preparative agarose gel. The cDNA pool was sequenced on an Illumina NextSeq 500 system using 1 × 150 bp read length. Library preparation and sequencing was performed by vertis Biotechnologie AG. The raw sequencing reads were demultiplexed and adapter trimming was performed with Cutadapt (v.4.1). Subsequently, we ran FastQC (v.0.11.8) for quality control. The reads were aligned and quantified with STAR (v.2.6.0a), using manually curated annotations for *Shigella flexneri* serotype 5a M90T based on NCBI accessions CP037923.1 and CP037924.1 (ref. 88). EdgeR (v.4.0.1) was used to analyze differential gene expression for each comparison[89]. Only genes with at least ten reads in at least three samples were included. Data were normalized using trimmed-mean of M-values (TMM) with default settings. log$_2$-FC was calculated with glmQLFit. Subsequently, a cutoff of ±1 was imposed on the log$_2$-FC and an FDR of 0.01 was selected to obtain a list of genes with significant differences in gene expression.

## In silico determination of bottleneck size

We estimated the size of the infection bottleneck from the pilot BO enteroid infections with sub-libraries containing 10,000 and 30,000 mutants each. The minimally enriched input was downsampled to infection bottleneck sizes between 0 and 6,000 bacteria with a stepsize of 100 (Fig. 1e), using the R base function sample. The median counts of the minimally enriched input libraries were used as weights. For every bottleneck size, we calculated the number of unique insertion sites. Comparing this number to the number of unique insertion sites in the output libraries (Fig. 1e), we estimated the size of the infection bottleneck to be about 2,000 bacteria. This results in 1,000 to 2,500 unique insertion sites per output library.

## In silico determination of number of replicates

To determine the number of sequencing libraries necessary to cover the *Shigella* genome based on the number of mutants per input sample, we conducted an in silico analysis based on the pilot data. Using the Minimal Library reference containing 130,000 *Shigella* mutants (including mutants in intergenic regions), we downsampled them to libraries with 1,000, 2,000 or 3,000 mutants each. Subsequently, the number of genes featuring at least three insertion sites across all libraries was calculated (Extended Data Fig. 3a) estimating that this threshold represents the minimum required for statistical significance. The analysis revealed that for library sizes ranging from 2,000 to 3,000 mutants, approximately 3,400 to 3,700 genes achieve a coverage of three insertion sites per gene when using 40 replicates. By contrast, achieving a comparable coverage with only 1,000 mutants per input sample would require more than 100 replicates. Additionally, the number of insertion sites per gene for 40 input libraries with 3,000 mutants each was calculated (Extended Data Fig. 3b). A total of 4,032 genes (83%, mean value across 100 runs) had at least one insertion site, 1,371 genes had 20 and more insertion sites. Consequently, we opted for 43 replicates with input library sizes between 2,000 and 3,000 mutants, anticipating that this would yield technical stochastic

mutant loss rates of approximately 50%, based on the number of unique insertion sites per output sample in the pilot infection assay (1,000–2,000) (Fig. 1d).

## Main TraDIS screen in BO enteroid infections

The *Shigella* Tn5 library 2 comprising 43 sub-libraries was used to infect 43 separate wells, each containing ~800 BO enteroids at MOI 40 for 6 h as described above. After infection, enteroids were lysed in 0.1% Na-deoxycholate by extensive pipetting. Intracellular bacterial populations were enriched ON at 30 °C in 2 ml LB broth. A diluted culture of the inoculum was also enriched and used as the input reference. A 1.5 ml portion of the bacterial culture was used to extract genomic DNA using the GenElute Bacterial gDNA kit. gDNA (1.5 µg) was cleaned with Ampure XP beads to remove EDTA. Cleaned gDNA (500 ng) was used as input for the sequencing library preparations. The NEXTFLEX Rapid XP library prep kit was used for fragmentation (300–400 bp), end repair and A-tailing. An Ampure XP bead clean-up was performed followed by splinkerette ligation as previously described[87]. Size selection was performed with the beads included in the NEXTFLEX Rapid XP kit. PCR amplification was performed as previously described[87] followed by a clean-up with the NEXTFLEX Cleanup Beads XP included in the kit. A single pool of the 43 input and the 43 output libraries was sequenced on a NextSeq 500 instrument over three runs using the 75-cycle High Output kit with a custom dark cycle recipe, similar to the recipe used on the above MiSeq runs, run as 1 × 60 bp.

## TraDIS read quantification, analysis and data preparation

The raw sequencing reads were demultiplexed and filtered by transposon tag allowing one mismatch. Sequencing adapters were removed with BBDuk (BBMap v.38.94) and quality control was performed using FastQC (v.0.11.8). The reads were aligned and quantified with the Bio-TraDIS pipeline (v.1.4.5), using manually curated annotations for *Shigella flexneri* serotype 5a M90T (NCBI accessions CP037923.1 and CP037924.1). The read counts for the insertion sites across all samples were concatenated and genomic positions without transposon insertion sites (TIS) removed using a custom Bash script. The unique insertion sites were assigned to genetic features and intergenic regions in R. For the TraDIS under different growth conditions, EdgeR[89] (v.4.0.1) was used to analyze significant changes in mutant fitness between the selected conditions. Only genes with at least ten reads in at least three samples were included. Data were normalized using TMM with default settings. $\log_2$-FC values were calculated with glmQLFit. Subsequently, a cutoff of 1 was imposed on the $\log_2$-transformed fold changes and an FDR of 0.01 was selected to obtain a list of genes with significant differences in gene expression. For the TraDIS screen in BO enteroid infections, insertion sites with fewer than 20 counts in the input sample were removed from the respective replicate. The input sample of sub-library 11 had only 300,000 aligned reads, about 5% of the average number of reads across all input samples (Fig. 2c) and was hence removed from subsequent analyses. This resulted in 85,464 nonzero TIS across 42 replicates, which were used as input to the Bayesian ZINB model.

## ZINB model for TraDIS screen

We developed a Bayesian model to extract genome-wide gene-wise fitness scores in the presence of experimental bottlenecks (Fig. 2d and Extended Data Fig. 3f). For a full model description including priors and determination of statistical significance, see Supplementary Note 1. In brief, we modeled the output counts for a single insertion site $Y_{rgk,out}$ with the ZINB distribution, where the probability of a stochastic (technical) zero due to experimental bottlenecks is given by the mixing coefficient $\theta_{rgk}$ and the expected mean $\mu_{rgk}$ depends on the corresponding input count $Y_{rgk,in}$, a gene-wise logFC $\log FC_g$ (fitness score) and a normalization constant $n_r$ between the input and output counts of an individual sub-library

$$\text{ZINB}(Y_{rgk,out}; \mu_{rgk}, \phi_{rgk}) = \begin{cases} \theta_{rgk} + (1 - \theta_{rgk})\, \text{NB}(0; 0, \phi_{rgk}), & \text{if } Y_{rgk,out} = 0, \\ (1 - \theta_{rgk})\, \text{NB}(Y_{rgk,out}; \mu_{rgk}, \phi_{rgk}), & \text{if } Y_{rgk,out} > 0. \end{cases}$$

$$(1)$$

We determined the technical parameters ($\theta_{rgk}$, $n_r$) by fitting the ZINB model to 12,573 insertion sites located within pseudogenes, assuming that mutations in these nonfunctional genes should not affect fitness, that is, $\log FC_g = 0$. Subsequently, we used these technical parameters to extract the fitness scores ($\log FC_g$) from 85,464 insertion sites with the full ZINB model.

To derive the posterior distributions of the model parameters, we used the probabilistic programming language Stan[90] (v.2.31.0). The statistical models were fitted to the TraDIS screen data running two chains of 1,000 Markov Chain Monte Carlo samples each. We assessed statistical significance and controlled the FDR by comparing $z$ values $z_g = \frac{\log FC_g}{\Delta \log FC_g}$ (with the posterior s.d. $\Delta \log FC_g$) to the standard normal distribution[42], assuming that at least 10% of genes have fitness effects during infection (Extended Data Fig. 3k,i).

## Immunostaining and confocal microscopy

After infection with *Shigella* p*uhpT*-GFP, BO or AO enteroids/colonoids were fixed with 2% paraformaldehyde for 30 min at room temperature and washed twice with PBS/0.25% BSA. Subsequently, enteroids/colonoids were permeabilized ON with 3% BSA, 0.1% saponin, followed by 4′,6-diamidino-2-phenylindole (DAPI; 1:1,000) (Sigma, D9542) and F-actin (1:400) (phalloidin-Alexa Fluor 647, Molecular Probes, A22287) staining for 1 h at room temperature. Samples were imaged at the BioVis platform of Uppsala University, using a LSM 700 AxioObserver (Zeiss) with a ×40/0.95 PlanApo objective, 405/488/639 diode lasers for excitation and the pinhole set to 1 a.u. for each channel.

## Mass spectrometry-based proteomics

The indicated wild-type *Shigella* and mutant strains were grown at 37 °C until $OD_{600}$ reached ~0.7. Bacterial cultures (2 ml) were spun down and washed once in PBS. Bacterial pellets were lysed in 50 µl 2% SDS and boiling at 98 °C for 10 min. Samples (5 µg) were denatured with a final concentration of 2% SDS and 20 mM Tris (2-carboxyethyl) phosphine. Samples were digested with a modified sp3 protocol[91] as previously described[92]. In brief, samples were added to a bead suspension (10 µg of beads (Sera-Mag Speed Beads, 4515-2105-050250 and 6515-2105-050250) in 10 µl 15% formic acid and 30 µl ethanol) and incubated shaking for 15 min at room temperature. Beads were then washed four times with 70% ethanol. Proteins were digested overnight by adding 40 µl of 5 mM chloroacetamide, 1.25 mM Tris (2-carboxyethyl) phosphine and 200 ng trypsin in 100 mM HEPES pH 8.5. Peptides were eluted from the beads and dried under vacuum. Peptides were then labeled with TMT-pro (ThermoFisher Scientific), pooled and desalted with solid-phase extraction using a Waters OASIS HLB µElution Plate (30 µm). Samples were fractionated onto 48 fractions on a reversed-phase C18 system running under high pH conditions, with every sixth fraction being pooled together. Samples were analyzed by liquid chromatography with tandem mass spectrometry, using a data-dependent acquisition strategy on a ThermoFisher Scientific Vanquish Neo LC coupled with a ThermoFisher Scientific Orbitrap Exploris 480. Raw files were processed with MSFragger[93] (v.3.0) against the NCBI *Shigella flexneri* 5a strain M90T genome (CP037923 and CP037924) using standard settings for TMT. Data were normalized using vsn[94] (v.3.74.0) and statistical significance was determined using limma[95] (v.3.62.2).

## Western blots

To evaluate IpaA-3xFT translation, equal amounts of total protein was extracted from the indicated strains following growth at 37 °C until $OD_{600}$ ~0.3 and induction by 0.25 mM IPTG for 30 min. Proteins were separated using Any kD Mini-PROTEAN TGX Stain-Free Protein

Gels (Bio-Rad, 4568126) and transferred onto Trans-Blot Turbo Mini 0.2 μm PVDF Transfer Packs (Bio-Rad, 1704156). Loading controls were obtained by the stain-free method, imaging each gel upon exposure to UV-light for 5 min[96]. Immunodetection was performed using an anti-Flag antibody (1:10,000) (Sigma, F1804), as described previously[97]. PageRuler Plus Prestained Protein Ladder (ThermoFisher Scientific, 26619) was used as weight marker. To quantify western blots, protein extracts were serially diluted and relative protein amounts calculated from a standard curve.

### Quantitative PCR with reverse transcription

Total RNA purification and cDNA synthesis were performed as previously described[98]. RT–qPCR was performed using Maxima SYBR green/ROX qPCR master mix (2×) (ThermoFisher Scientific, K0222) on a CFX384 Touch Real-Time PCR Detection System (Bio-Rad). The levels of *ipaA*, *mnmE* and *mnmG* transcripts were analyzed using the $2^{-\Delta\Delta Ct}$ (cycle threshold) method and results are reported as the fold increase relative to the reference. The housekeeping gene *nusA* was used for normalization. Oligonucleotide primers used are listed in Supplementary Table 3.

### Codon usage analysis

Codon usage was analyzed using a Python (v.3.9.16) script and the BioPython (v.1.81) library. In brief, the occurrences of each codon for each nonpseudogene coding sequence was counted on the chromosome and virulence plasmid (accessions CP037923.1 and CP037924). The relative frequencies for each synonymous codon were calculated globally and for each nonpseudogene coding sequence.

### Statistics and reproducibility

Sample sizes were not predetermined, except for the Main TraDIS screen in BO enteroid infections (43 biological replicate sub-libraries; see above). For the main TraDIS screen in BO enteroid infections, sub-library 11 was excluded due to low sequencing read counts. TraDIS pilot experiments in BO enteroids were done in biological triplicate. TraDIS under different growth conditions and RNA-seq experiments were performed in biological triplicate. Proteomic profiling was done using four biological replicates per strain. Barcoded assays were performed at least in duplicate (in this setup each replicate used two biological replicates for each genotype, allowing for a powerful and internally controlled comparison). All other experiments were performed with at least three biological independent replicates. Exact sample sizes indicated in each figure legend. All experiments were highly reproducible and our sample sizes were similar to previous related publications. The experiments were not randomized. Data collection and analysis was not blinded. Statistical comparisons were done with two-sided *t*-tests, or two-sided Mann–Whitney *U*-test, as described in each figure legend.

### Reporting summary

Further information on research design is available in the Nature Portfolio Reporting Summary linked to this article.

### Data availability

Transcriptomic data and TraDIS data have been deposited in the Gene Expression Omnibus (GEO) database, with the SuperSeries no. GSE267520. The mass spectrometry proteomics data have been deposited at the ProteomeXchange Consortium via the PRIDE partner repository with the dataset identifier PXD046629. Source data are provided with this paper.

### Code availability

For TraDIS under different growth conditions we used the Bio-TraDIS pipeline[87] (v.1.4.5) to align and quantify reads and EdgeR[89] (v.4.0.1) to analyze significant changes in fitness. For differential gene expression we used STAR (v.2.6.0a) and EdgeR (v.4.0.1). For mass spectrometry-based proteomics MSFragger was used[93], data were normalized using vsn[94] and statistical significance was determined using limma[95]. For the main TraDIS screen in BO enteroid infections we developed a custom Bayesian ZINB model (Supplementary Note 1; Methods). For codon usage analysis we used a custom Python (v.3.9.16) script and the BioPython (v.1.81) library. All custom code was deposited to Zenodo at the following links: https://doi.org/10.5281/zenodo.15096674 (ref. 99) and https://zenodo.org/records/15100662 (ref. 100).

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

## Acknowledgements

We are grateful to members of the Sellin laboratory and to J. Näsvall for helpful discussions, and to the staff at our surgical units for assistance with human intestinal tissue sampling for enteroid and colonoid establishment. We acknowledge the Ramaciotti Center for Genomics, Sydney, Australia for TraDIS library preparation and

sequencing. This work was supported by grants from ESCMID (Research Grant 2020 to M.L.D.M.), the Carl Trygger Foundation (CTS 22:1915 to M.L.D.M.), the Clas Groschinsky Memorial Foundation (M21112 to M.L.D.M.), the Bavarian State Ministry for Science and the Arts through the research network bayresq.net (to L.B.), the Swedish Research Council (2018-02223, 2022-01590 to M.E.S.), the Swedish Foundation for Strategic Research (FFL18-0165 to M.E.S.) and the SciLifeLab Fellows program (to M.E.S.). The proteomics analysis was enabled by a grant from Kempestiftelserna (JCK3126 to A.M.). M.P. acknowledges support by a 6-month international visiting scholarship from Sapienza University of Rome.

## Author contributions

Conceptualization: M.L.D.M., M.E.S., L.J. and L.B. Validation: M.L.D.M. and M.E.S. Investigation: M.L.D.M., A.B., J.E., A.N., A.C.C.L., M.P. and A.M. Methodology: M.L.D.M., L.J., A.B., M.P., A.M., L.B. and M.E.S. Formal analysis: M.L.D.M., L.J., A.B., J.E. and A.M. Resources: M.L.D.M., M. Sundbom, M. Skogar, W.G., D.-W.L., P.M.H., A.M., LB. and M.E.S. Software: L.J., J.E. and L.B. Data curation: M.L.D.M. Visualization: M.L.D.M., L.J. Project administration: M.L.D.M., M. Sundbom, M. Skogar, W.G., D.-L.W., P.M.H. and M.E.S. Supervision: M.L.D.M., L.B. and M.E.S. Funding acquisition: M.L.D.M., L.B. and M.E.S. Writing—original draft: M.L.D.M., L.J., L.B. and M.E.S. Writing–review and editing: all authors.

## Funding

## Competing interests

The authors declare no competing interests.

## Additional information

**Extended data** is available for this paper at https://doi.org/10.1038/s41588-025-02218-x.

**Correspondence and requests for materials** should be addressed to Maria Letizia Di Martino, Laura Jenniches, Lars Barquist or Mikael E. Sellin.

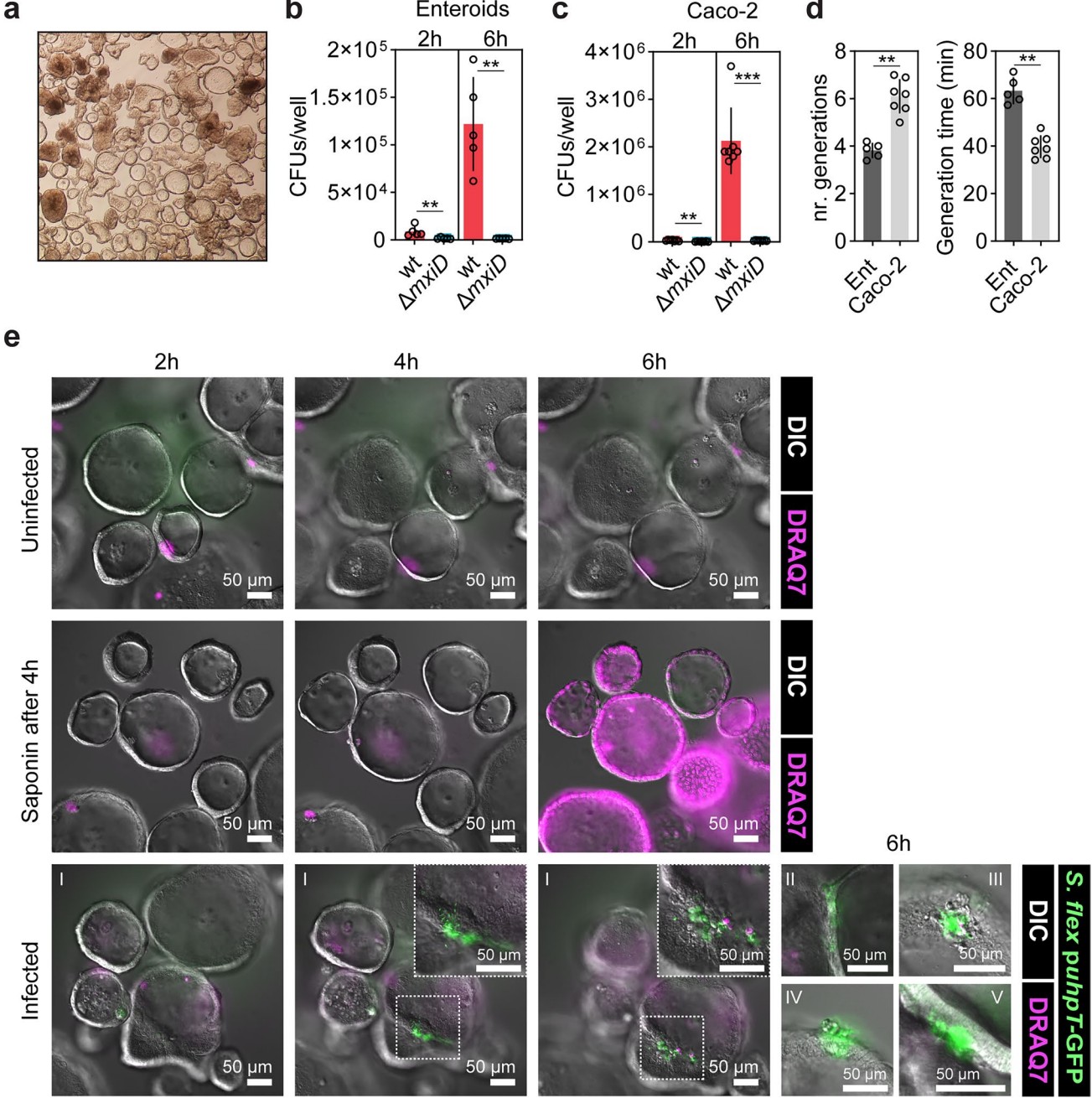

**Extended Data Fig. 1 | Characteristics of the basal-out enteroid infection model for early *Shigella* colonization of human intestinal epithelium.** (Extended Data Fig) Representative image showing a BO enteroid suspension culture as it appears before infection. (**b**) *Shigella* CFU counts upon coinfection of BO enteroids with a mix of wt and Δ*mxiD* (non-invasive) strains for 2 h and 6 h at MOI 40. CFU counts come from 5 biological replicates pooled from 2 independent experiments. Data shown as Mean +/- SD. Significance determined by two-sided Mann-Whitney U-test; **p < 0.01. (**c**) *Shigella* CFU counts upon coinfection of Caco-2 cells with a mix of wt and Δ*mxiD* (non-invasive) strains for 2 h and 6 h at MOI 40. CFU counts come from 7 biological replicates pooled from 2 independent experiments. Data shown as Mean +/- SD. Significance determined by two-sided Mann-Whitney U-test; **p < 0.01; ***p < 0.001. (**d**) Graph showing

the number of generations (left panel) and the generation time (right panel) for *Shigella* intracellular populations in BO enteroids and Caco-2 cells between 2 and 6 h pi. Number of generations was calculated using CFUs counts from Extended Data Fig. 1b-c. Data shown as Mean +/- SD. Significance determined by two-sided Mann-Whitney U-test; **p < 0.01. (**e**) Representative time-lapse series of BO enteroids stained with DRAQ7 and treated as following: uninfected (first row); uninfected with addition of saponin after 4 h (second row) and, infected with *Shigella* wt containing the cytosolic reporter p*uhpT*-GFP at MOI 40 (third row). Experiments were repeated 4 times with imaging of at least 10 enteroids/ experiment. Roman numbers represent five different examples of infection foci. Scale bar: 50 μm.

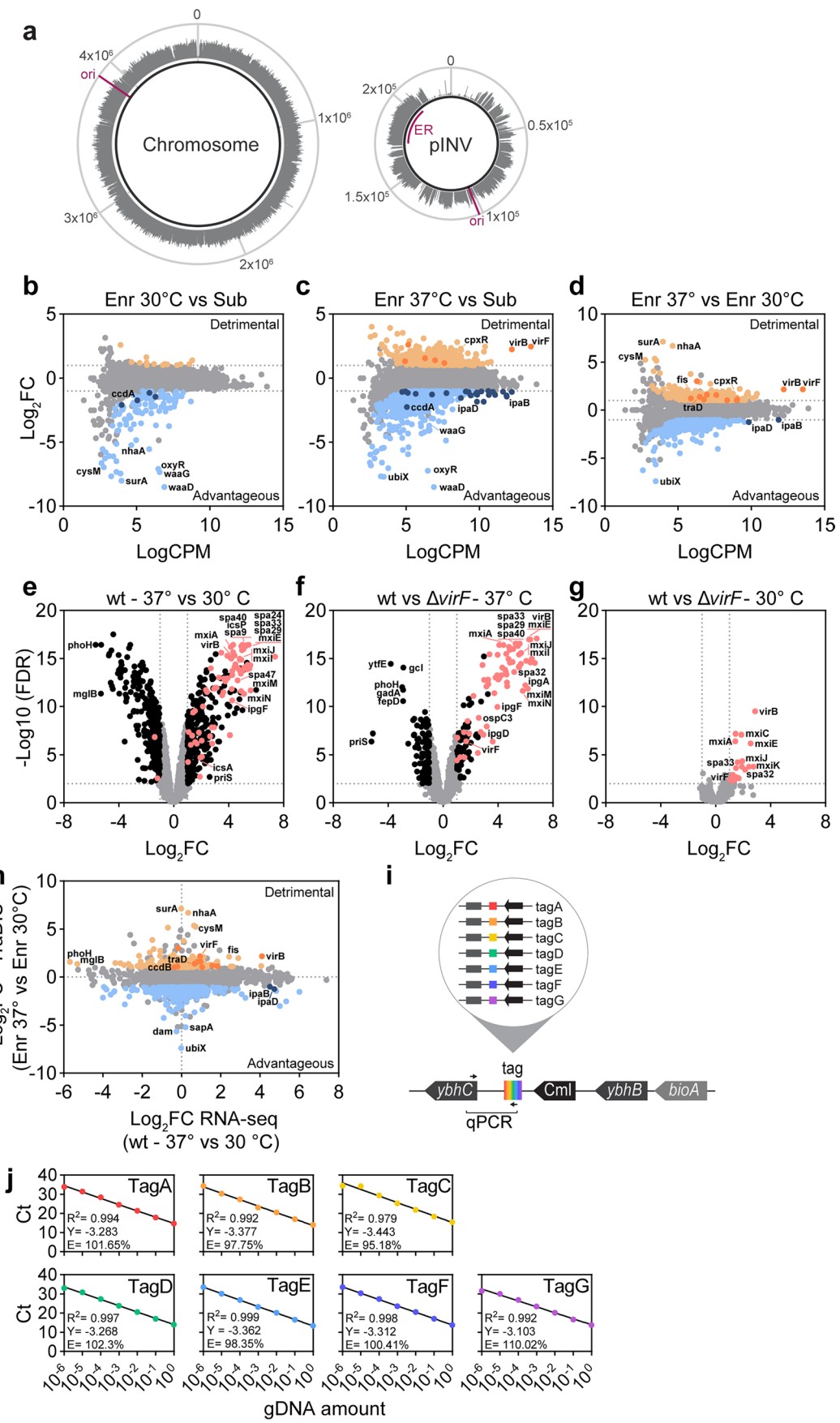

**Extended Data Fig. 2 | See next page for caption.**

**Extended Data Fig. 2 | Optimization of experimental design for combining TraDIS with *Shigella* enteroid infections.** (**a**) *Shigella* M90T chromosome and pINV maps showing the TIS distribution across the Tn5 Library #1 (grey). Origin of replication (ori) for chromosome and pINV and entry region (ER) for pINV in magenta. (**b–d**) MA plots of relative mutant fitness expressed as Log$_2$FC between Enr 30 °C vs Sub (**b**), Enr 37 °C vs Sub (**c**) and Enr 37 °C vs Enr 30 °C (**d**). Each dot represents a gene. Detrimental genes located on pINV shown in dark orange, detrimental genes on the chromosome in light orange, advantageous genes on the pINV in dark blue, and advantageous genes on the chromosome in light blue. Significance determined by two-sided F-test with Benjamini-Hochberg correction for multiple hypothesis testing. Log$_2$FC ≥ 1; FDR ≤ 0.01. (**e–g**) Volcano plots showing differentially expressed genes in *Shigella* wt grown at 37 °C vs 30 °C (**e**), *Shigella* wt vs Δ*virF* mutant grown at 37 °C (**f**) and *Shigella* wt vs Δ*virF* mutant grown at 30 °C (**g**). Each dot represents a gene. Differentially expressed genes located on the pINV shown in pink, on the chromosome in black, and all non-significant differentially expressed genes in grey. Significance determined

by two-sided F-test with Benjamini-Hochberg correction for multiple hypothesis testing. Log$_2$FC ≥ 1; FDR ≤ 0.01. (**h**) Plot showing relative mutant fitness for both pINV- and chromosome-located genes on the y-axis (Log$_2$FC as noted in Extended Data Fig. 2d – Enr 37 °C vs Enr 30 °C) versus the respective expression changes on the x-axis (RNA-Seq – Log$_2$FC as noted in Extended Data Fig. 2e for *Shigella* wt grown at 37 °C vs 30 °C). Significant pINV-encoded advantageous (dark blue) or detrimental (dark orange) and chromosomal advantageous (light blue) or detrimental (light orange) genes are shown. Log$_2$FC ≥ 1; FDR ≤ 0.01. (**i**) Schematic representation of the tag-containing *Shigella* chromosomal locus. Primer binding sites for qPCR tag quantification indicated. (**j**) Standard curves for qPCR detection of tags A-G, using specific primer pairs (see Supplementary Table 3). gDNA from pure monocultures of each of seven *Shigella* wt tagA-G barcoded strains were diluted (ten-fold dilutions up to 10$^{-6}$) and used to generate standard curves for each primer pair. R$^2$, slope (Y) and efficiency (**E**) values indicated in each panel.

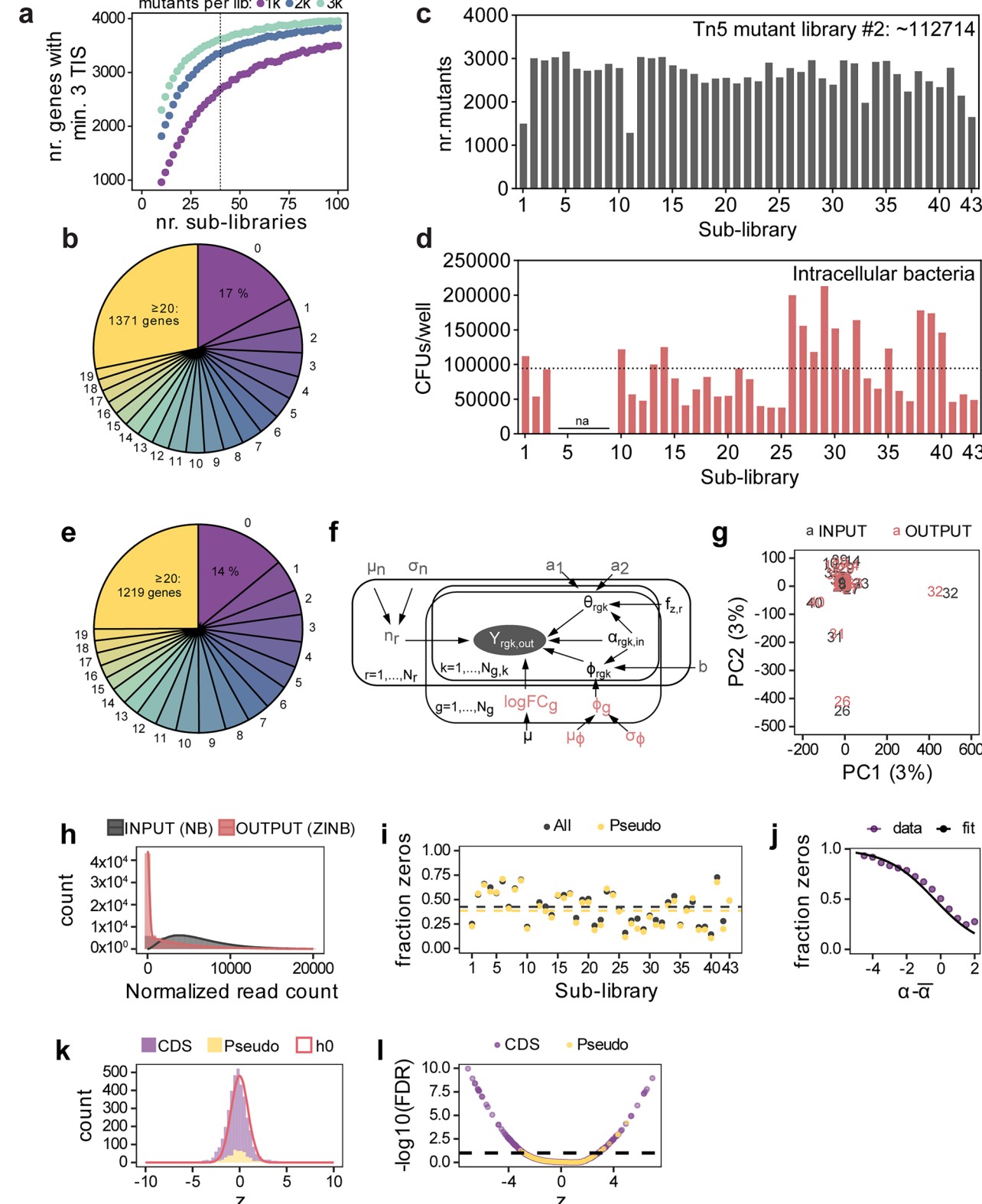

**Extended Data Fig. 3 | See next page for caption.**

**Extended Data Fig. 3 | *Shigella* mutant library characteristics and modelling p rameters. (a)** *In silico* estimation of the number of genes with at least 3 TIS for input sub-libraries with 1,000-3,000 mutants. **(b)** Pie chart showing expected gene-wise TIS number across input sub-libraries of 40 replicates with 3,000 mutants each, based on the simulation in Extended Data Fig. 3a. 17% of genes had no TIS, ~1,371 ≥ 20 TIS. **(c)** The *Shigella* Tn5 mutant library #2 for enteroid infections pooled into 43 sub-libraries. Shown is the number of mutants/sub-library determined by CFU plating. **(d)** Total *Shigella* intracellular population sizes upon enteroid infection with sub-libraries in c, determined by CFU plating. No data exist for sub-libraries 4-9. **(e)** Pie chart showing the gene-wise TIS number across samples of the 42 *Shigella* Tn5 mutant input sub-libraries. Sub-library 11 removed due to low input sequencing read counts (see Fig. 2c). 14% of genes had no TIS, ~1,219 genes ≥20 TIS. **(f)** Plate diagram depicting the ZINB model to map *Shigella* invasion factors (see also Fig. 2d). Technical parameters (normalization $n_r$, dependence of mutant loss on abundance $\alpha_{rgk,in}$ and fraction of zeros in the output sample $f_{z,r}$) estimated from pseudogenes in grey, gene-fitness scores ($logFC_g$) and disperson ($\Phi_g$) estimated from all genes in red. **(g)** PCA plot of TraDIS data from the *Shigella* enteroid infections. Input-output sub-libraries cluster by replicate. **(h)** Histogram showing that distribution of read counts of input TIS can be approximated by a negative binomial distribution (red line). Infection bottleneck leads to zero-inflated count distribution in output samples. **(i)** Fraction of TIS with zero counts in output (and non-zero counts in the respective input) varies between ~25 and ~75% across sub-libraries. **(j)** Fraction of zero counts in the output depends on the input TIS abundance. Y-axis shows fraction of zero counts, x-axis TIS input abundance normalized by average TIS input abundance. **(k)** Histogram showing the distribution of z-values approximated by a normal distribution around zero with a standard deviation of 0.9 (see Methods, Supplementary Note 1). **(l)** The FDR corresponds to the cumulative fraction of z-values above and below the normal distribution (see Methods). Dashed line indicates an FDR cut-off of 0.01.

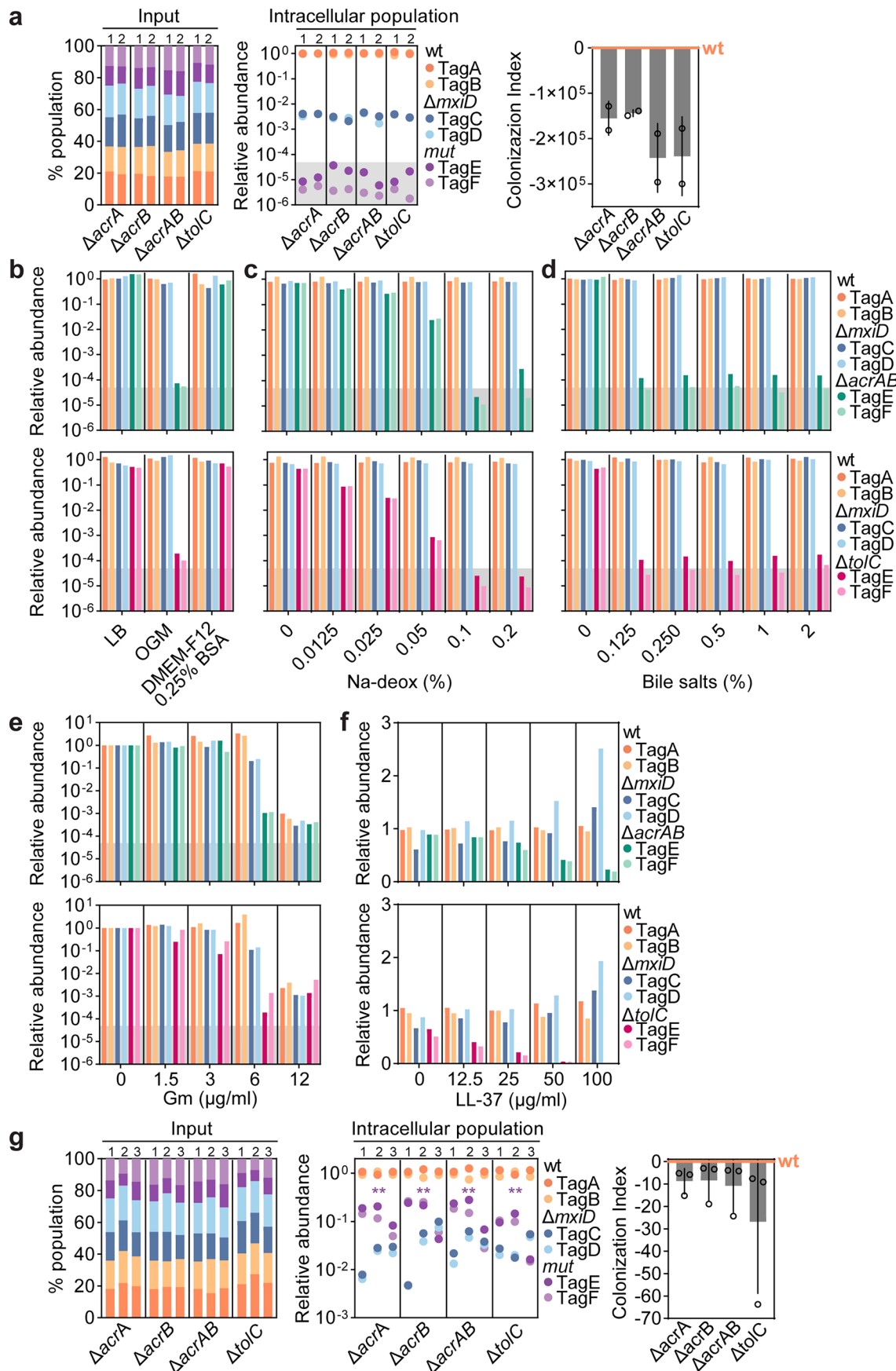

**Extended Data Fig. 4 | See next page for caption.**

**Extended Data Fig. 4 | Contribution of the AcrABTolC efflux pump system to *Shigella* colonization of enteroids, and resilience to environmental conditions relevant to the infection assay.** (**a**) BO enteroids were infected with barcoded consortia comprising two *Shigella* wt (tagA, tagB), two Δ*mxiD* (tagC, tagD) and two mutant (tagE, tagF) strains at MOI 40 for 6 h in Organoid growth medium (OGM). Mut refers to mutant specified on x-axis. Left panel, percentage of each strain in the input population. Middle panel, relative tag abundance in the intracellular population, normalized against the corresponding input (see Left panel). Right panel, Colonization Index for the indicated mutants (derived from data in the middle panel and calculated as 1-(wt/mut)). Data in right panel shown as Mean +/- SD. Data for two independently generated consortia per infection. Grey shading indicates detection limit. Note that relative abundances of Δ*acrA*, Δ*acrB*, Δ*acrAB*, and Δ*tolC* mutants fall under the detection limit and are ~1000-fold lower than the non-invasive Δ*mxiD* strain. (**b**–**f**) Barcoded competition assay with a consortium comprising two wt (tagA, tagB), two Δ*mxiD* (tagC, tagD) and two Δ*acrAB* or Δ*tolC* mutant (tagF, tagG) strains grown in different media (LB broth, Organoid growth medium (OGM) or DMEM-F12/0.25%BSA) (**b**), or LB broth with increasing concentration of Na-deoxycholate (**c**), Bile salts (**d**), gentamicin (**e**), or LL-37 (**f**). Shown is the relative tag abundance under each condition. Relative tag abundances normalized to the consortia grown in LB. (**g**) BO enteroids infected with barcoded consortia as in a, but in DMEM-F12/0.25%BSA medium. Data for three independently generated consortia per infection. Significance determined by two-sided paired t-test between normalized output and input abundances (see Methods). **$p < 0.01$. Data in right panel shown as Mean +/- SD. <u>Interpretation:</u> Δ*acrA*, Δ*acrB*, Δ*acrAB* and Δ*tolC* mutants were not only sensitive to bile salts/deoxycholate, but also to the OGM medium used for infections (see b-d). Therefore, the low $Log_2FC$ value observed for these mutants in the TraDIS screen does not exclusively reflect decreased enteroid colonization, but also generalized sensitivity to multiple infection condition stressors (**a**–**f**). Nevertheless, an epithelial colonization defect was validated in the absence of OGM and Na-deoxycholate. Here, the Colonization Indexes for Δ*acrA*, Δ*acrB*, Δ*acrAB* mutants were ~10-fold lower and for Δ*tolC* mutant ~20-30-fold lower than the wt (g; compare to a).

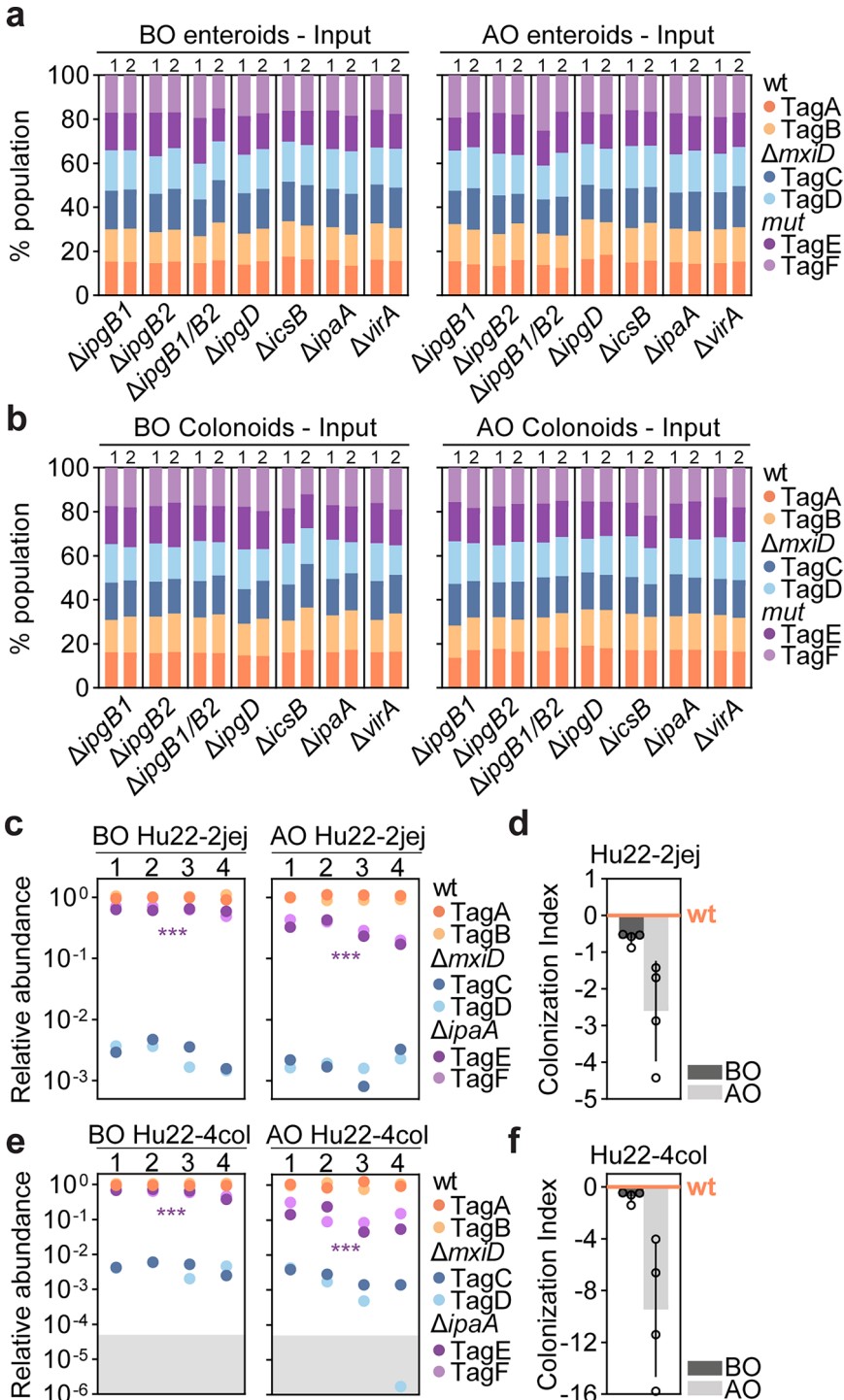

**Extended Data Fig. 5 | Input compositions and phenotypic validations for barcoded consortium infections with strains carrying mutations in *Shigella* T3SS effecto s.** (**a, b**) BO and AO enteroids (**a**) and BO and AO colonoids (**b**) were infected with a mixed barcoded consortium comprising two wt (tagA, tagB), two Δ*mxiD* (tagC, tagD) and two mutant (tagE, tagF) strains at MOI 40 for 6 h. Mut refers to the specific mutant as indicated on x-axis. Graphs depict the composition of barcoded *Shigella* consortia used as input inoculums for experiments in Fig. 3j–m. The relative abundance of each tag in the input consortia is plotted as percentage of the total population. Note that no strain is consistently over- or underrepresented. Shown are data for two independently generated consortia for each infection. See Fig. 3j–m for the corresponding relative tag abundances in the intracellular population and Colonization Index scores. (**c–f**) BO and AO enteroids (ID 22-2jej) (**c**) and BO and AO colonoids (ID

22-4col) (**e**) were infected with a mixed barcoded consortium comprising two wt (tagA, tagB), two Δ*mxiD* (tagC, tagD) and two Δ*ipaA* (tagE, tagF) strains at MOI 40 for 6 h. Shown is the quantification of relative tag abundance, normalized against the corresponding input. For each geometry, infections were executed with four independently generated consortia. Significance determined by two-sided paired t-test for each specific mutant between the normalized output and the normalized input abundances (see Methods). ***p < 0.001. (**d–f**) Bar graph showing the Colonization Index for each mutant in BO (dark grey) and AO (light grey) enteroid 22-2jej infections (derived from data in c and calculated as 1-(wt/mut); sample size as in c) and BO (dark grey) and AO (light grey) colonoid 22-4col infections (derived from data in e and calculated as 1-(wt/mut); sample size as in e). wt is shown as 0. Data shown as Mean +/- SD.

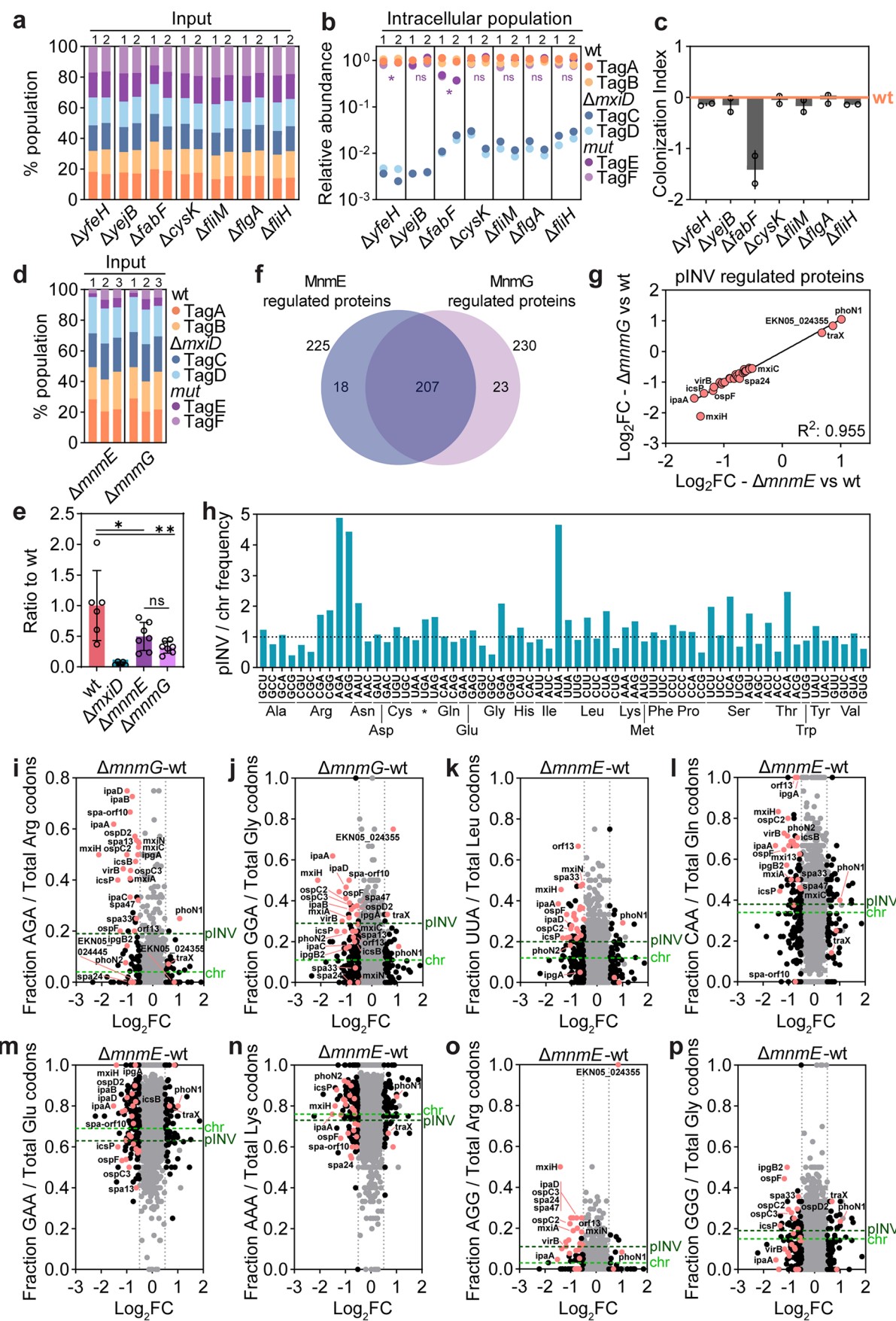

**Extended Data Fig. 6 | See next page for caption.**

**Extended Data Fig. 6 | Enteroid colonization phenotypes for chromosomal** *Shigella* **genes, and analysis of codon usage b.** (**a**–**c**) BO enteroids infected with barcoded consortia comprising two wt (tagA, tagB), two Δ*mxiD* (tagC, tagD) and two mutant (tagE, tagF) strains at MOI 40 for 6 h. Mut refers to the mutant specified on x-axis. (**a**) Percentage of each strain in the input and (**b**) relative tag abundance in the intracellular population, normalized against the corresponding input. Data for two independent consortia per infection. Significance determined by two-sided paired t-test (see Methods). ns - non-significant; *p < 0.05. (**c**) Colonization Indexes for the indicated mutants (Derived from b and calculated as 1-(wt/mut)). Data shown as Mean +/- SD. (**d**) Percentage of each strain in the input for barcoded consortia used for infections in Fig. 4b. (**e**) Intracellular bacteria in BO enteroids infected with *Shigella* wt, Δ*mxiD*, Δ*mnmE* or Δ*mnmG* for 3 h at MOI 40. CFU counts from 6 (wt), 5 (Δ*mxiD*) and 7 (Δ*mnmE*/ Δ*mnmG*) biological replicates pooled from 2 independent experiments, normalized to wt. Data shown as Mean +/− SD. Significance determined by two-tailed Mann-Whitney U-test; ns - non-significant; *p < 0.05; **p < 0.01. (**f**)

Venn-diagram of *Shigella* differentially expressed proteins in Δ*mnmE* vs wt (MnmE regulated proteins), and Δ*mnmG* vs wt (MnmG regulated proteins); Log$_2$FC ≥ 0.5, adj_pvalue ≤ 0.01. (**g**) Linear regression between pINV differentially expressed proteins in Δ*mnmE* vs wt (MnmE regulated proteins), and Δ*mnmG* vs wt (MnmG regulated proteins); Log$_2$FC ≥ 0.5, adj_pvalue ≤ 0.01. (**h**) Codon usage frequency in pINV-located *Shigella* genes compared to chromosomal genes. Dashed line indicates no enrichment. (**i, j**) Relative protein levels (Log$_2$FC as in Fig. 4d – Δ*mnmG* vs wt) versus AGA codon usage ratios for Arg (**i**) and GGA for Gly (**j**) per protein. (**k**–**p**) Relative protein levels (Log$_2$FC as in Fig. 4c – Δ*mnmE* vs wt) versus UUA codon usage ratios for Leu (**k**), CAA for Gln (**l**), GAA for Glu (**m**), AAA for Lys (**n**), AGG for Arg (**o**) and GGG for Gly (**p**) per protein. For i-p, differentially expressed proteins on pINV shown in pink, on chromosome in black, and all non-significant differentially expressed proteins in grey. Log$_2$FC ≥ 0.5; adj_pvalue ≤ 0.01. Dark and light green dashed lines specify mean codon usage for all open reading frames on pINV, or chromosome, respectively.

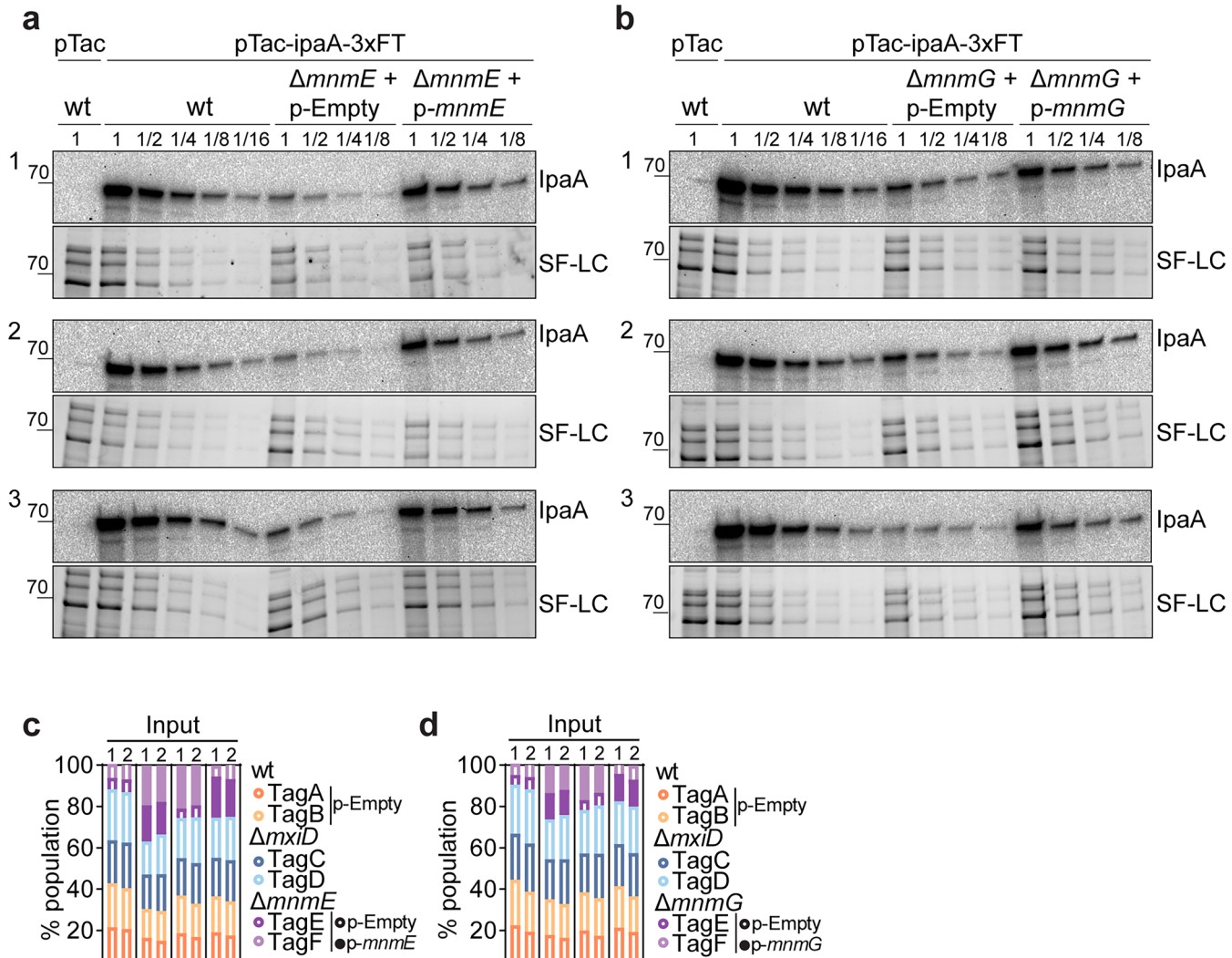

**Extended Data Fig. 7 | Effects of *mnmE* and *mnmG* complementation on IpaA-3xFT protein translation and *Shigella* fitness during inoculum growth.** (**a**) Relative IpaA-3xFT protein content in *Shigella* wt and Δ*mnmE* strains carrying an inducible pTac-ipaA-3xFT plasmid and the p-Empty or p-*mnmE* plasmids, determined by quantification of western blots of serially diluted samples. Shown are 3 independent western blots of serially diluted samples for each strain, used for generating standard curves and calculating the relative protein abundance. See Fig. 4j for the corresponding IpaA-3xFT protein content quantification. (**b**) Relative IpaA-3xFT protein content in *Shigella* wt and Δ*mnmG* strains carrying an inducible pTac-ipaA-3xFT plasmid and the p-Empty or p-*mnmG* plasmids, determined by quantification of western blots of serially diluted samples. Shown are 3 independent western blots of serially diluted samples for each strain, used

for generating standard curves and calculating the relative protein abundance. See Fig. 4j for the corresponding IpaA-3xFT protein content quantification. (**c**) BO enteroids were infected with mixed barcoded consortia comprising the indicated strains at MOI 40 for 6 h, as shown in Fig. 4k. The relative abundance of each tag in the input consortia is plotted as percentage of the total population. Shown are data for two independently generated consortia for each infection. (**d**) BO enteroids were infected with mixed barcoded consortia comprising the indicated strains at MOI 40 for 6 h, as shown in Fig. 4l. The relative abundance of each tag in the input consortia is plotted as percentage of the total input population. Shown are data for two independently generated consortia for each infection.

# Reporting Summary

## Statistics

For all statistical analyses, confirm that the following items are present in the figure legend, table legend, main text, or Methods section.

| n/a | Confirmed | |
|---|---|---|
| ☐ | ☒ | The exact sample size (*n*) for each experimental group/condition, given as a discrete number and unit of measurement |
| ☐ | ☒ | A statement on whether measurements were taken from distinct samples or whether the same sample was measured repeatedly |
| ☐ | ☒ | The statistical test(s) used AND whether they are one- or two-sided<br>*Only common tests should be described solely by name; describe more complex techniques in the Methods section.* |
| ☒ | ☐ | A description of all covariates tested |
| ☐ | ☒ | A description of any assumptions or corrections, such as tests of normality and adjustment for multiple comparisons |
| ☐ | ☒ | A full description of the statistical parameters including central tendency (e.g. means) or other basic estimates (e.g. regression coefficient) AND variation (e.g. standard deviation) or associated estimates of uncertainty (e.g. confidence intervals) |
| ☐ | ☒ | For null hypothesis testing, the test statistic (e.g. $F$, $t$, $r$) with confidence intervals, effect sizes, degrees of freedom and $P$ value noted<br>*Give P values as exact values whenever suitable.* |
| ☐ | ☒ | For Bayesian analysis, information on the choice of priors and Markov chain Monte Carlo settings |
| ☒ | ☐ | For hierarchical and complex designs, identification of the appropriate level for tests and full reporting of outcomes |
| ☒ | ☐ | Estimates of effect sizes (e.g. Cohen's *d*, Pearson's *r*), indicating how they were calculated |

*Our web collection on statistics for biologists contains articles on many of the points above.*

## Software and code

Policy information about availability of computer code

| Data collection | No software was used for data collection. |
|---|---|
| Data analysis | - For TraDIS under different growth conditions we used the Bio-TraDIS pipeline (v1.4.5) to align and quantify reads and EdgeR (v4.0.1) to analyze significant changes in fitness.<br>- For differential gene expression we used STAR (v2.6.0a) and EdgeR (v4.0.1) (Robinson et al 2010).<br>- For the main TraDIS screen in basal-out enteroid infections we used a custom Bayesian ZINB model, available through the following Zenodo link: https://doi.org/10.5281/zenodo.15096674.<br>- For Mass spectrometry-based proteomics MSFragger (v3.0) was used (Kong et al, 2017). Data was normalized using vsn (v3.74.0) (Huber et al, 2002) and statistical significance was determined using limma (v3.62.2) (Ritchie et al, 2015).<br>- For codon usage analysis we used a Python (3.9.16) script and the BioPython (1.81) library. All code is open source and available at https://github.com/Oftatkofta/codon_counter and through the following Zenodo link: https://zenodo.org/records/15100662. |

For manuscripts utilizing custom algorithms or software that are central to the research but not yet described in published literature, software must be made available to editors and reviewers. We strongly encourage code deposition in a community repository (e.g. GitHub). See the Nature Portfolio guidelines for submitting code & software for further information.

## Data

Policy information about availability of data

All manuscripts must include a data availability statement. This statement should provide the following information, where applicable:
- Accession codes, unique identifiers, or web links for publicly available datasets
- A description of any restrictions on data availability
- For clinical datasets or third party data, please ensure that the statement adheres to our policy

- Transcriptomic data and TraDIS data have been deposited in the Gene Expression Omnibus (GEO) database, with the SuperSeries no. GSE267520, link: https://www.ncbi.nlm.nih.gov/geo/query/acc.cgi?acc=GSE267520.

- The mass spectrometry proteomics data have been deposited at the ProteomeXchange Consortium via the PRIDE partner repository with the dataset identifier PXD046629, link: https://www.ebi.ac.uk/pride/archive/projects/PXD046629.

## Research involving human participants, their data, or biological material

Policy information about studies with human participants or human data. See also policy information about sex, gender (identity/presentation), and sexual orientation and race, ethnicity and racism.

| | |
|---|---|
| Reporting on sex and gender | Not relevant to our study. |
| Reporting on race, ethnicity, or other socially relevant groupings | Not relevant to our study. |
| Population characteristics | Not relevant to our study. No data on population characteristics was collected/used. |
| Recruitment | Human adult stem cell-derived enteroids/colonoids were established from jejunal tissue re-sected during bariatric surgery (enteroids), or from morphologically normal non-tumor colon tissue resected during elective colon cancer surgery (colonoids), in all cases following the subject's informed consent. To ensure the anonymity of tissue donors, samples were pseudonymized. Patients' identities were not accessible to laboratory personnel. |
| Ethics oversight | The procedures were approved by the local governing body (Etikprövningsmyndigheten, Sweden, Sweden) under license nr 2010-157 with addenda 2010-157-1 and 2020-05754, and license nr 2023-01524-01. |

Note that full information on the approval of the study protocol must also be provided in the manuscript.

# Field-specific reporting

Please select the one below that is the best fit for your research. If you are not sure, read the appropriate sections before making your selection.

☒ Life sciences ☐ Behavioural & social sciences ☐ Ecological, evolutionary & environmental sciences

For a reference copy of the document with all sections, see nature.com/documents/nr-reporting-summary-flat.pdf

# Life sciences study design

All studies must disclose on these points even when the disclosure is negative.

| | |
|---|---|
| Sample size | - TraDIS experiments under different growth conditions were done in biological triplicates.<br>- TraDIS pilot experiments in basal-out enteroids were done in biological triplicates.<br>- RNA-seq experiments were done in biological triplicates.<br>- Main TraDIS screen in basal-out enteroid infections was done using 43 biological replicate sub-libraries. Sample size was predetermined experimentally ans in silico as detailed in the methods (see "TraDIS pilot screen in basal-out enteroids" and "In silico determination of number of replicates").<br>- Proteomic profiling was done using four biological replicates per strain.<br>- Barcoded assays were performed at least in duplicates (In this setup each replicate employed two biological replicates for each genotype, allowing for a powerful and internally controlled comparison).<br>- All other experiments were performed with at least 3 biological independent replicates. Exact sample sizes are indicated in each figure legend.<br>Our replicate numbers were consistent with those reported in similar studies within the field, indicating that our experimental design was appropriately aligned with established standards. Additionally, the validation of the TraDIS results through multiple approaches, along with the high reproducibility of all experiments, indicates that our sample sizes were sufficient. |
| Data exclusions | For the main TraDIS screen in basal-out enteroid infections, sub-library 11 was excluded due to low sequencing read counts. |
| Replication | Experimental replicates are indicated in the figure legends, and each experiment was repeated at least two times. |

| Randomization | Not relevant to our study since all bacterial strains/samples were treated in the same way. |
| Blinding | Not relevant to our study since all bacterial strains/samples were treated in the same way. |

# Reporting for specific materials, systems and methods

We require information from authors about some types of materials, experimental systems and methods used in many studies. Here, indicate whether each material, system or method listed is relevant to your study. If you are not sure if a list item applies to your research, read the appropriate section before selecting a response.

## Materials & experimental systems

| n/a | Involved in the study |
|---|---|
| ☐ | ☒ Antibodies |
| ☐ | ☒ Eukaryotic cell lines |
| ☒ | ☐ Palaeontology and archaeology |
| ☒ | ☐ Animals and other organisms |
| ☒ | ☐ Clinical data |
| ☒ | ☐ Dual use research of concern |
| ☒ | ☐ Plants |

## Methods

| n/a | Involved in the study |
|---|---|
| ☒ | ☐ ChIP-seq |
| ☒ | ☐ Flow cytometry |
| ☒ | ☐ MRI-based neuroimaging |

## Antibodies

| Antibodies used | Anti-FLAG antibody (Sigma, #F1804) for Western Blots. |
| Validation | Anti-FLAG antibody has been validated by Sigma, as specified in the product description, accessible through the homepage of the manufacturer. From Sigma: "We have employed an affinity resin to purify an ANTI-FLAG M2 monoclonal antibody exhibiting excellent specificity and high sensitivity. This affinity-purified ANTI-FLAG M2 antibody has been utilized to detect tagged fusion proteins in multiple expression systems, displaying virtually exclusive selectivity for the target protein band in Western blot immunostaining."<br>To further confirm the selectivity of the antibody in our system, we included a negative control sample which does not express the FLAG-tagged protein of interest, in all western blots. |

## Eukaryotic cell lines

Policy information about cell lines and Sex and Gender in Research

| Cell line source(s) | - Human adult stem cell-derived enteroids/colonoids were established from jejunal tissue resected during bariatric surgery (enteroids), or from morphologically normal non-tumor colon tissue resected during elective colon cancer surgery (colonoids), in all cases following the subject's informed consent.<br>- Caco-2 cells (ATCC HTB-37) |
| Authentication | Eurofins authenticated Caco-2 cells using STR profiling. Caco-2 cells were continuously monitored for their well-defined phenotypic behavior including capacity to form a high-TEER monolayer and microvilliated cell morphology atop PET transwell inserts. |
| Mycoplasma contamination | - Enteroids and Colonoid cultures were not tested for mycoplasma contamination. Sentinel cultures from the cell culture facility were however tested for Mycoplasma contamination and were found to be negative during the entire duration of this study.<br>- Caco-2 cells were confirmed to be mycoplasma-free by PCR following growth in culture medium without antibiotics. |
| Commonly misidentified lines (See ICLAC register) | No commonly misidentified cell lines were used in the study. |

## Plants

Seed stocks

*Report on the source of all seed stocks or other plant material used. If applicable, state the seed stock centre and catalogue number. If plant specimens were collected from the field, describe the collection location, date and sampling procedures.*

Novel plant genotypes

*Describe the methods by which all novel plant genotypes were produced. This includes those generated by transgenic approaches, gene editing, chemical/radiation-based mutagenesis and hybridization. For transgenic lines, describe the transformation method, the number of independent lines analyzed and the generation upon which experiments were performed. For gene-edited lines, describe the editor used, the endogenous sequence targeted for editing, the targeting guide RNA sequence (if applicable) and how the editor was applied.*

Authentication

*Describe any authentication procedures for each seed stock used or novel genotype generated. Describe any experiments used to assess the effect of a mutation and, where applicable, how potential secondary effects (e.g. second site T-DNA insertions, mosiacism, off-target gene editing) were examined.*

