## [Peer Review File · Nature Genetics]

A Scalable Gut Epithelial Organoid Model Reveals the Genome-Wide Colonization Landscape of a Human-Adapted Pathogen

Corresponding Author: Professor Mikael Sellin

Version 0:

Decision Letter:

17th Sep 2024

Dear Professor Sellin,

First, I am so sorry for the delay in returning this decision to you. Thank you for your patience.

Your Article, "A Scalable Gut Epithelial Organoid Model Reveals the Genome-Wide Colonization Landscape of a Human Restricted Pathogen" has now been seen by 3 referees. You will see from their comments below that while they find your work of interest, some important points are raised. We are interested in the possibility of publishing your study in Nature Genetics, but would like to consider your response to these concerns in the form of a revised manuscript before we make a final decision on publication.

We therefore invite you to revise your manuscript taking into account all reviewer and editor comments. Please highlight all changes in the manuscript text file. At this stage we will need you to upload a copy of the manuscript in MS Word .docx or similar editable format.

*2) If you have not done so already please begin to revise your manuscript so that it conforms to our Article format instructions, available

[here](http://www.nature.com/ng/authors/article_types/index.html).

*3) Include a revised version of any required Reporting Summary: <https://www.nature.com/documents/nr-reporting-summary.pdf>

Please be aware of our [guidelines](https://www.nature.com/nature-research/editorial-policies/image-integrity) on digital image standards.

Link Redacted

Note: This URL links to your confidential home page and associated information about manuscripts you may have

submitted, or that you are reviewing for us. If you wish to forward this email to co-authors, please delete the link to your homepage.

We hope to receive your revised manuscript within four to eight weeks. If you cannot send it within this time, please let us know.

Sincerely,

Safia Danovi, PhD
Senior Editor, Nature Genetics
ORCID: 0009-0007-7822-5479

Referee expertise:

Referee #1: modelling Shigella infections

Referee #2: intestinal co-cultures

Referee #3: Shigella/host interactions

Reviewers' Comments:

Reviewer #1:

Remarks to the Author:

Using large scale Shigella infection of enteroids and colonoids in combination with TraDIS, computational modelling, barcoded consortium infections and proteomics, Di Martino et al provide a first genome wide map of Shigella genes required to colonize human intestinal epithelium. The authors define a Shigella 'invasome' (including 143 genes) and a 'minimal set' of T3SS effectors required for epithelial invasion across intestinal segments and geometries (including IpgB1, IpgB2, IpaA, VirA). They also reveal a novel mechanism of global virulence regulation mediated by tRNA modification enzymes MnmE/G and differential codon usage. The work provides a inspirational framework to advance genome wide screens in organoids and other complex infection models.

The overall scale and depth of work is impressive and exciting. In addition to its clinical relevance, Shigella is a paradigm for discovery in microbiology, cell biology and immunology. This work can inspire a wide variety of researchers. I have comments to help this work reach its full potential across a broad readership.

1. The authors develop a scalable human intestinal epithelium model for Shigella infection (Figure 1). Clever testing is performed at different temperatures (30°C, 37°C) to link bacterial fitness with T3SS virulence gene expression, expanding previous gene essentiality studies (the Shigella field has for the most part been strictly focused on working at 37°C). Have authors tested T3SS protein secretion at 30°C vs 37°C? Can infection of organoids be performed at 30°C?

2. The authors provide a genome wide map of the Shigella 'infectome' required to colonize human intestinal epithelium (Figure 2). Overall results are novel, identifying chromosomal and pINV genes for future work to mechanistically pursue.
-The work provides 86% coverage of annotated genes. What about non annotated genes?
-Mutants in LPS and O antigen biosynthesis reduce invasiveness. Future work should compare with Shigella sonnei considering its relevance to global health and literature describing its reduced invasiveness and unique O antigen.
-Analysis revealed a beneficial nonredundant effect a select set of T3SS effectors (IpgB1, IpgD, IcsB, IpaA, VirA). Considering the deep literature on Shigella effectors, and their role in evasion of cell-autonomous immunity, I am most surprised by these results. Why does this approach fail to detect more Shigella effectors previously reported to be required for virulence (OspC, OspF, OspG, IpaH1.4 etc)?

3. The authors discover a minimal set of T3SS effectors which promote Shigella invasion of human epithelia across intestinal segments and geometries (Figure 3), helping to resolve conflicting results in the field from groups using different infection models.
- IpgB1, IpgB2, IpaA, VirA- what is role of these effectors? How do the authors view this a minimal set? More discussion is required.

-What is driving context dependence of IpaA? At least speculate and help readers to understand.

4. The authors refer to their work as testing 'early' infection / colonisation. What is meant by early (replication is tested at many hours post infection)? *Shigella* is viewed to escape from the vacuole within 15 minutes (from work using HeLa cells). At what stage of invasion is affected by replication defects: cell entry, cytosolic replication, other? In this way, testing at multiple different timepoints would be valuable to differentiate.

5. It may be surprising that actin-based motility is not captured here (no discussion of IcsA/VirG). The authors look at CFU as a primary read out. This work would be more broadly appreciated if also testing for bacterial dissemination and activation of host defence processes (including autophagy, cell death) for which *Shigella* is widely recognised to induce.

6. Organoid infection is *in vitro* and results are likely to be different when other immune cells are introduced (eg macrophages, neutrophils). Beyond new genes lists, can the authors more clearly state how this work can lead us to therapeutics or vaccine candidates (translational advance). What is future of this work? Mechanistic testing of candidate gene lists? Using clinically relevant *Shigella* isolates? Manipulation of host factors? Would results change +/- IFN γ ?

7. Contextualise these screening results. Can the authors compare data sets with previously available data sets using other models (beyond Table S4)? Compare *Shigella* interactome with other human adapted pathogens?

8. Briefly discuss limitations of this study. What about genes not called in this work under these conditions? Are they not important?

Reviewer #2:

Remarks to the Author:

Di Martino, Jenniches et al. present a genome-wide screen of invasion factors in *Shigella flexneri*. Using an elegant combination of Transposon Directed Insertion Sequencing in *S. flexneri* and organoid-based invasion studies, the authors identify more than 100 genes linked to bacterial invasion into healthy human gut epithelial cells. The identification of 2 tRNA modification enzymes that regulate invasion-linked virulence gene expression represents an unexpected finding of this study that is convincingly linked to invasion modulation. Overall, this study is performed to a very high standard with numerous steps of technical establishment to ensure appropriate identification of stochastic mutant losses. I expect it to serve as a gold standard for similar future studies.

The main area where I see much untapped potential and some weaknesses is the comparative use of different organoid lines and protocols. The following points could substantially strengthen this manuscript:

1. If I interpret the methods section correctly, one jejunal and one colonic organoid line from different patients were used throughout this study. While the comparative statements between GI tract regions are highly interesting, the variation between these 2 patients is a striking confounding factor for any such statements in the current setup. Running the entire TraDIS screen in more organoid lines may be excessive, but validation of the key KO strains in at least 2 more organoid lines per intestinal segment (ideally paired from the same patient, but I understand that this may not be feasible) would be necessary to substantiate these statements.

2. In Figure 3 K/M, few patterns of apical-basal preference seem to be preserved. I have concerns about the robustness of this assay, these should be resolved by further replicates and ideally more organoid lines (see 1). The potential divergence (or neutrality) between apical and basal invasion is an intriguing aspect that can be studied very well with the platform developed by the authors, so that more robust statements on this aspect will further strengthen the manuscript.

3. The Gentamicin protection assay used throughout this study may be validated in greater depth (or supported by further literature references). How were the concentrations of 200 μ g/ml for 2h or 20 μ g/ml for 6h p.i. determined, and are those sufficient to kill all bacteria of this strain in the absence of organoids cells?

Beyond these points, I think the study is exceptionally well-done, robust, interesting and warrants publication.

Reviewer #3:

Remarks to the Author:

The manuscript by Di Martino and coworkers describes the application of TraDIS to identify genes relevant for *Shigella flexneri* infection of gut epithelial organoid models. The manuscript depicts the optimization of the organoids as a *Shigella* infection model amenable to the TraDIS large-scale approach. Through this approach, both chromosomally and plasmid-encoded genes that modulate the infection were identified, including a minimal set of T3SS effectors and two tRNA modifying enzymes.

Overall, this study describes an elegant methodology combining TraDIS and organoid models to tackle an important area of research, specifically *Shigella* infection. This is pertinent given the lack of relevant animal models to study infection by this bacterial pathogen. The manuscript results constitute a useful resource for the entire community.

However, while the manuscript is interesting from a methodology and resource standpoint, the follow-up experiments on selected hits do not provide novel biology/mechanistic insights.

All the comments below are aimed at a clarification of the methodology and dataset.

1. Although I agree with the authors that there are severe limitations regarding animal models recapitulating Shigella infection in humans and that organoids are, in principle, more biologically relevant than cancer cell lines, the strong emphasis on this point should be toned down. Along this line, it would be highly relevant to discuss also the limitations inherent to the use of organoid models in the context of infection, as well of their applicability to high-throughput approaches.

2. The authors mention throughout the manuscript that the genes identified are essential for epithelial cell invasion. The use of the term invasion is rather surprising and not accurate, given that the TraDIS and most validation experiments were performed at 6 hpi. As such, the identified factors could affect (likely are) other steps of the interaction of the bacteria with epithelial cells (e.g. bacterial replication and/or clearance).

3. Regarding the description of TraDIS, specifically the optimized protocol used for screening of the full library, the authors describe the application to the 43 sub-libraries as replicates of infection (Fig 2A and text). The use of replicates in this context does not appear accurate.

4. Why was the infection/sequencing for the sub-library 11 not repeated?

5. What is the overlap between the Δ mmE/WT and Δ mmG/WT differentially expressed proteins? Are the two tRNA modification enzymes functionally redundant? It would be interesting to add mechanistic and biological relevance for the data provided.

Version 1:

Decision Letter:

Our ref: NG-A66137R

4th Mar 2025

Dear Dr Sellin,

Thank you for submitting your revised manuscript "A Scalable Gut Epithelial Organoid Model Reveals the Genome-Wide Colonization Landscape of a Human Restricted Pathogen" (NG-A66137R). It has now been seen by Reviewers #1,#3 and their comments are below. Please note that Reviewer #1 also provided feedback on your response to Reviewer #2 who did not return a report.

The reviewers find that the paper has improved in revision, and therefore we'll be happy in principle to publish it in Nature Genetics, pending minor revisions to satisfy our editorial and formatting guidelines.

Sincerely,

Safia Danovi, PhD
Senior Editor, Nature Genetics
ORCID: 0009-0007-7822-5479

Reviewer #1 (Remarks to the Author):

This is an exciting time for Shigella infection biology. Novel, reliable infection models are desperately needed. In this revised manuscript authors carefully consider all reviewer comments, and fully address the breadth of reviews with additional experiments and text updates.

I have no further comments and look forward to the impact this work will have on the infection biology community.

FEEDBACK ON AUTHOR RESPONSE TO REVIEWER #2

I continue to enjoy thinking about the manuscript and its impact.

Reviewer2 is positive: '...elegant combination..', '...convincingly linked..', '...study performed to a very high standard...', '...serve as a gold standard...'. '...Beyond these points, I think the study is exceptionally well done, robust, interesting and warrants publication.'

Reviewer2 comment 1, 2 (linked) about untapped potential / weakness: As acknowledged by the authors, this comment is highly relevant and results would be more convincing if reproduced across further donor cultures. To address this authors advance barcoded consortium infections in additional enteroid and colonoid lines. New results confirm context independent requirement for mxiD and context dependent requirement for lpaA. Importantly, consistency between biological replicates and between experiments conducted in separate enteroid / colloid lines is compelling.

Reviewer2 comment 3 about gentamicin. Additional experiments were performed, clearly showing that gentamicin concentrations used in enteroid / colonoid infections eradicate extracellular Shigella.

Together, I feel authors have carefully considered and addressed comments from all reviewers (including reviewer2) and the manuscript is significantly strengthened as a result.

Reviewer #3 (Remarks to the Author):

The authors have done extensive work and have introduced changes to the manuscript to address the main comments of all reviewers, and the revised version of the manuscript is significantly improved. As mentioned previously, I find that this manuscript provides an important and comprehensive resource for researchers working on Shigella infections.

We would like to thank all three reviewers for their positive evaluations and constructive suggestions on our manuscript NG-A66137 entitled “A Scalable Gut Epithelial Organoid Model Reveals the Genome-Wide Colonization Landscape of a Human Restricted Pathogen”. We have incorporated these suggestions and responded point-by-point to each comment below. The original reviewer comments are included in *italicized* text, followed by our replies. In addition to clarifications in writing, we have conducted several additional experiments, guided by the reviewer comments. This has led to the incorporation of new experimental data and analyses in the revised manuscript, specifically:

I. A new supplemental Figure to more thoroughly characterize the enteroid model for *Shigella* infection used in the TraDIS screen (FigS1, new data included in panels b-e).

II. New barcoded consortium infection data performed in basal-out (BO) as well as apical-out (AO) enteroid and colonoid cultures, established from additional tissue donors (new data included in FigS5, panels c-f).

III. A comparison between differentially abundant proteins in the $\Delta mnmE$ vs $\Delta mnmG$ *Shigella* mutant strains (new analyses included in FigS6, panels f-g).

Beyond this, further control experiments and analyses have also been conducted, which we have deemed informative, but not essential to include in the revised manuscript itself. Such results are presented exclusively here in the response letter. Taken together, these revision experiments and analyses have furthered strengthened the conclusions of our study and provided important context. We have adapted the manuscript text accordingly, as specified below. Please note that the figure call outs and line numbers reported in this response letter refer to the track changed version of the revised manuscript.

REVIEWER #1 (modelling Shigella infections)

Using large scale Shigella infection of enteroids and colonoids in combination with TraDIS, computational modelling, barcoded consortium infections and proteomics, Di Martino et al provide a first genome wide map of Shigella genes required to colonize human intestinal epithelium. The authors define a Shigella ‘invasome’ (including 143 genes) and a ‘minimal set’ of T3SS effectors required for epithelial invasion across intestinal segments and geometries (including lpgB1, lpgB2, lpaA, VirA). They also reveal a novel mechanism of global virulence regulation mediated by tRNA modification enzymes MnmE/G and differential codon usage. The work provides a inspirational framework to advance genome wide screens in organoids and other complex infection models.

The overall scale and depth of work is impressive and exciting. In addition to its clinical relevance, Shigella is a paradigm for discovery in microbiology, cell biology and immunology. This work can inspire a wide variety of researchers. I have comments to help this work reach its full potential across a broad readership.

Response: We are happy to learn of this positive overall assessment. Below we specify how the remaining concerns of reviewer #1 have been addressed.

1. The authors develop a scalable human intestinal epithelium model for *Shigella* infection (Figure 1). Clever testing is performed at different temperatures (30°C, 37°C) to link bacterial fitness with T3SS virulence gene expression, expanding previous gene essentiality studies (the *Shigella* field has for the most part been strictly focused on working at 37°C). Have authors tested T3SS protein secretion at 30°C vs 37°C? Can infection of organoids be performed at 30°C?

Response: *Shigella* coordinates virulence factor expression, mainly located on the large virulence plasmid pINV, through complex regulatory networks to colonize and disseminate in the host gut epithelium. Prior work has established that at 30°C virulence gene expression is minimal, mainly explained by the transcriptional silencing of the master regulator VirF that drives expression of the T3SS and its effectors (see e.g. ¹⁻³). This can also be observed in our RNA-Seq dataset in which virulence gene expression is strongly upregulated at 37°C vs 30°C in a *Shigella* wt strain (FigS2e - differentially expressed genes located on the pINV shown in pink). Nevertheless, we have here substantiated the impact of *Shigella* growth temperature on its ability to invade intestinal epithelial cells under the current conditions. For this purpose, we have infected human enteroids with a mix of wt and $\Delta mxiD$ (non-invasive; lacking a structural component of the T3SS) strains, with strains sub-cultured either at 30°C or 37°C prior to infection. We performed infections at MOI 40. At 6h p.i. we could retrieve ~90,000 CFUs/infection for the wt *Shigella* strain when the inoculum was pre-grown at 37°C, but only ~500 CFUs (i.e. ~0.6% of the 37°C counts) for the wt *Shigella* strain when grown at 30°C. Importantly, we retrieved similar CFUs for the wt *Shigella* strain grown at 30°C as for the $\Delta mxiD$ strain grown at either 30°C or 37°C. This indicates that wt *Shigella* grown at 30°C behaves as a strain devoid of the main T3SS invasion machinery (FigR1 – only shown in this response letter). As these findings are as expected from ample previous literature, we suggest to not include these new data in the revised manuscript itself. However, we are prepared to reconsider, if the reviewer and editor find this an important premise to specifically emphasize.

FigR1. *Shigella* CFU counts upon coinfection of BO enteroids with a mix of wt and $\Delta mxiD$ (non-invasive) strains sub-cultured at 30°C or 37°C at MOI 40. Infections were carried out for 40 min before adding Gentamicin. Enteroids were lysed at 6h p.i. and intracellular bacterial populations were serially diluted and plated on LB plates containing appropriate antibiotics. CFU counts come from 3 biological replicates and each symbol represents a paired replicate.

2. The authors provide a genome wide map of the *Shigella* ‘infectome’ required to colonize human intestinal epithelium (Figure 2). Overall results are novel, identifying chromosomal and pINV genes for future work to mechanistically pursue.

-The work provides 86% coverage of annotated genes. What about non annotated genes?

Response: The gene annotations we used are based on a recent assembly annotated with state-of-the-art bacterial gene annotation tools and cover 89% of the genome sequence⁴. Importantly, the gene annotations also include e.g. hypothetical coding sequences without a known functional annotation. In terms of unannotated genes, this will largely consist of small non-coding RNAs (ncRNAs) and short open reading frames (sORFs). To explore the potential of our data to interrogate these elements, we have compared the transposon insertion site frequency in the remaining intergenic regions to annotated genes (FigR2 – only shown in this

response letter). This analysis shows that the insertion frequency in intergenic regions is higher compared to most annotated genes, suggesting that the majority of these insertions are neutral, at least in the conditions of library construction. This analysis also suggests there may be sufficient insertion density to assay at least some of these unannotated elements, given appropriate annotations. Unfortunately, reliable annotation of ncRNAs and sORFs remains challenging, and often relies on a combination of experimental and computational approaches together with manual annotation. This would constitute a research project in itself, which we leave to future work.

FigR2. Distribution of transposon insertion site (TIS) frequency in annotated genes for chromosome and plasmid, calculated as the number of TIS per gene divided by its length. Red lines indicate the TIS frequency in the intergenic regions.

-Mutants in LPS and O antigen biosynthesis reduce invasiveness. Future work should compare with Shigella sonnei considering its relevance to global health and literature describing its reduced invasiveness and unique O antigen.

Response: Several mutants linked to LPS and O antigen biosynthesis show a positive Log_2FC in the output libraries (intracellular population) compared to the input libraries. This indicates an increased invasiveness, which is likely explained by the fact that these mutants harbour a shorter LPS, favouring the T3SS docking into the IEC membrane. However, LPS-affecting mutants are also known to be attenuated *in vivo* due to poor cell envelope protection (see^{5,6}). A deeper comparison between *Shigella flexneri* and *Shigella sonnei* indeed represents an exciting suggestion for future work along these lines. In revision, we have adjusted the description of the results relating to LPS biosynthesis and included two literature references that describe relevant LPS-related infection phenotypes for *Shigella sonnei* (^{7,8}) (Lines 202-206).

-Analysis revealed a beneficial nonredundant effect a select set of T3SS effectors (ipgB1, ipgD, icsB, ipaA, virA). Considering the deep literature on Shigella effectors, and their role in evasion of cell-autonomous immunity, I am most surprised by these results. Why does this approach fail to detect more Shigella effectors previously reported to be required for virulence (OspC, OspF, OspG, IpaH1.4 etc)?

Response: After rereading the relevant manuscript passages with this comment in mind, we agree that this point requires clarification. The TraDIS screen we have conducted gives a snapshot of what happens up until 6h p.i. in the enteroid model. We have here optimized the assay conditions to strike a balance between a large enough intracellular *Shigella* population and minimizing the impact of epithelial cell death. To further substantiate the impact of cell death in this infection time frame, we have now infected the enteroids with wild-type (wt) *Shigella* harbouring the intracellular reporter *puhpT*-GFP in the presence of DRAQ7, a membrane-impermeable dye that stains the DNA of dead or permeabilized cells. By 4h p.i. *Shigella* infection foci started to appear in IECs. Between 4 and 6h p.i., foci increased in size and early cell-to-cell spread was also evident. Importantly, very little pyroptotic cell death was observed up until 6h p.i., compared with a control sample in which saponin was added to the enteroids after 4h (new FigS1e). Hence, due to the relatively short infection window of the screen, it is not surprising that effectors specifically involved in innate immunity evasion and

suppression of cell death were not picked up as strong hits. For example, we have shown in a recent pre-print that the *Shigella* OspC3 effector is indeed important to evade innate immune recognition and thereby spread laterally in enteroid and colonoid-derived monolayers, which becomes evident at later infection time points (see⁹). The data below have been included in the revised manuscript, combined with appropriate clarifications in the results and discussion text (Lines 100-102 and 334-337).

FigS1 – Panel e. (e) Representative time-lapse series of BO enteroids kept in medium containing DRAQ7 and treated as following: uninfected (first row), uninfected with addition of saponin after 4h (second row) and, infected with the *Shigella* wt strain harbouring the intracellular reporter *puhpT-GFP* at MOI 40 (third row). Roman numerals represent five different examples of infection foci in these movies. Scale bar: 50 μ m.

3. The authors discover a minimal set of T3SS effectors which promote *Shigella* invasion of human epithelia across intestinal segments and geometries (Figure 3), helping to resolve conflicting results in the field from groups using different infection models.

- *IpgB1, IpgB2, IpaA, VirA*- what is role of these effectors? How do the authors view this a minimal set? More discussion is required.

Response: In response to this comment, we have added appropriate clarifications of the molecular actions of these T3SS effectors in the results text (Lines 260-264) and also included further discussion to contextualize these observations (Lines 366-369).

-What is driving context dependence of *IpaA*? At least speculate and help readers to understand.

Response: The deeper molecular basis for this apical preference is at this stage unclear. For example, *IpaA* has been demonstrated to bind Vinculin (see^{10,11}), which could be relevant for a differential impact of this effector during basal versus apical entry into IECs. As advised, we have included a speculation along these lines in the discussion (Lines 371-372).

4. The authors refer to their work as testing 'early' infection / colonisation. What is meant by early (replication is tested at many hours post infection)? *Shigella* is viewed to escape from the vacuole within 15 minutes (from work using HeLa cells). At what stage of invasion is affected by replication defects: cell entry, cytosolic replication, other? In this way, testing at multiple different timepoints would be valuable to differentiate.

Response: Thank you for pointing this out (as also commented on by Reviewer #3). To better clarify these aspects, we have added a new supplemental figure (FigS1), in which we characterized the enteroid *Shigella* infection model further. To compare enteroid infections with commonly used infection models for *Shigella* (e.g. Caco-2 cells), we have performed comparative co-infections with a mix of wt and $\Delta mx i D$ (non-invasive; lacking a structural component of the T3SS) strains at MOI 40 over different time points. It appears evident that intracellular population sizes are noticeably smaller in the enteroid model compared to Caco-2 cells, and that this is already observed in the initial time window (FigS1b-c, 2h p.i. wt - Ent: 8,931 CFUs/well ; wt - Caco-2: 31,857 CFUs/well). This can be explained by a generalizable trend of non-transformed epithelial cells to be more challenging to invade for enteric pathogens, as we have recently shown for *Salmonella* Typhimurium (see ^{12,13}). However, here we also found that *Shigella* intracellular replication between 2 and 6h p.i. is restricted to a greater extent in human enteroids compared to in Caco-2 cells. This results in a lower number of generations and a higher generation time for the intracellular bacterial population (FigS1d; 2-6h p.i. time window). As shown above, we also observed limited lytic cell death in *Shigella* infected enteroids up until 6h p.i. (FigS1e). Finally, we found several mutants for metabolic processes and transporters to be less capable of colonizing the enteroids, while mutants for known immune evasion- and cell death-suppressing effectors did not show obvious colonization defects within this relatively short infection window (as elaborated on under comment 2 above). From this, we conclude that the *Shigella* populations assayed in the TraDIS screen at 6h p.i. arise by i) active T3SS-driven invasion, followed by ii) ~4 generations of intraepithelial expansion, iii) in the presence of only minimal levels of epithelial cell death. We have included these clarifying new data in the revised manuscript, and revised the text accordingly (Lines 100-102). We have also pruned the phrasing to consistently use the word "colonization" throughout the manuscript.

FigS1 - Panels b-d. (b) *Shigella* CFU counts upon coinfection of BO enteroids with a mix of wt and $\Delta mx i D$ (non-invasive) strains for 2h and 6h at MOI 40. CFU counts come from 5 biological replicates pooled from 2 independent experiments. Statistical significance determined by Mann-Whitney U-test; **p<0.01.

(c) *Shigella* CFU counts upon coinfection of Caco-2 cells with a mix of wt and $\Delta mx i D$ (non-invasive) strains for 2h and 6h at MOI 40. CFU counts come from 7 biological replicates pooled from 2 independent experiments. Statistical significance determined by Mann-Whitney U-test; **p<0.01; ***p<0.001.

(d) Graphs showing the deduced number of generations (left panel) and the generation time (right panel) for *Shigella* intracellular population in BO enteroids and Caco-2 cells between 2 and 6h pi. These values were calculated using the CFU counts from Fig S1b-c. Statistical significance determined by Mann-Whitney U-test; **p<0.01.

5. It may be surprising that actin-based motility is not captured here (no discussion of *IcsA/VirG*). The authors look at CFU as a primary read out. This work would be more broadly appreciated if also testing for bacterial dissemination and activation of host defence processes (including autophagy, cell death) for which *Shigella* is widely recognised to induce.

Response: We acknowledge that we did not discuss this aspect in the initially submitted manuscript. However, we actually did find actin-based motility to be relevant for enteroid colonization within the present time frame (up to 6h p.i.); the *icsA* mutant is modestly, but significantly, less able to colonize enteroids in the TraDIS screen (Log_2FC : -0.63; FDR: 2.06E-17; Fig2f; TableS3). As we reason also above, it is highly likely that prolonging the infection time would result in a progressively larger impact of actin-based motility (as well as innate immune defense evasion strategies) for enteroid colonization. Indeed, we have shown in a recent pre-print using enteroid- and colonoid-derived monolayers that intracellular spread is severely impacted for a ΔicsA mutant between 8 and 20h p.i. (see⁹). We have added a sentence in the revised manuscript to highlight this aspect (Lines 210-211). Finally, screens to test the genome-wide requirements for *Shigella* dissemination and host defence processes at longer time scales represent exciting suggestions for future studies.

6. Organoid infection is *in vitro* and results are likely to be different when other immune cells are introduced (eg macrophages, neutrophils). Beyond new genes lists, can the authors more clearly state how this work can lead us to therapeutics or vaccine candidates (translational advance). What is future of this work? Mechanistic testing of candidate gene lists? Using clinically relevant *Shigella* isolates? Manipulation of host factors? Would results change +/- IFNg?

Response: This point is well taken. In response to this reviewer comment, we have in the revised manuscript discussion described potential trajectories, as well as current limitations, of the approach (Lines 334-340; 398). Due to the short format of Nature Genetics articles, these extensions have by necessity been kept brief. The impact of IFNg priming on *Shigella* spread within human enteroids and colonoids is a topic we are in fact gearing up to study by live-cell imaging in the near future.

7. Contextualise these screening results. Can the authors compare data sets with previously available data sets using other models (beyond Table S4)? Compare *Shigella* interactome with other human adapted pathogens?

Response: The present work is to our knowledge the first example where a 3D organotypic infection model has been successfully employed to probe the colonization capacity of a human-specific pathogen in a genome-wide fashion. For *Shigella* infections, there has also been a shortage of animal models that permit such genetic screens. Hence, there is a limitation in available data sets to compare our screening results to in a meaningful way.

Nevertheless, we have below attempted to compare decreased/increased mutant fitness between our *Shigella* enteroid infection screen results and a Tn-seq screen of human-restricted *Salmonella* Typhi and Paratyphi A infecting human macrophages (see¹⁴). The conditions of that other study included two passages of macrophage infection at MOI 100, for 18h each, using *Salmonella* Paratyphi A or *Salmonella* Typhi with a ΔVi capsule background to increase the frequency of intracellular bacteria, and with no bacterial population enrichment subsequent to the infection. First, we used Proteinortho to define orthologous genes between the *Shigella* genome and the two *Salmonella* Tn-seq genomes. This resulted in ~1,000 orthologous genes. To visualize the comparisons, we generated Venn diagrams (*Shigella*: FDR<0.2, *Salmonella*: FDR<0.05, and $|\log_2\text{FC}|>0.5$) for mutants showing either increased-, or decreased-, fitness during *Shigella* enteroid infection. This analysis showed that there is no overlap of genes with positive fitness changes (mutants better able to colonize) between the *Shigella* and *Salmonella* screens, and only a very limited overlap between genes with negative

fitness changes (mutants less able to colonize) (FigR3 – only shown in this response letter). Considering i) that *Shigella* and *Salmonella*, although related, utilize different strategies to colonize their hosts (as we have illustrated in a recent pre-print; see⁹), ii) that human intestinal epithelial enteroids/colonoids and human macrophages represent distinct infection environments, iii) that the infection setup also differs markedly, and iv) in light of the stringent word limit of Nature Genetics articles, we have decided not to add this analysis into the revised manuscript.

FigR3. a. Venn-diagram of orthologous genes showing increased fitness during infection in *Shigella flexneri* (enteroid infections, TraDIS screen in this manuscript), *Salmonella Typhi* and *S. Paratyphi A* (macrophage infections; see¹⁴). (*Shigella*: FDR<0.2, *Salmonella*: FDR<0.05, and $|\log_2FC|>0.5$). b. Venn-diagram of orthologous genes showing decreased fitness during infection in *Shigella flexneri* (enteroid infections, TraDIS screen in this manuscript), *Salmonella Typhi* and *S. Paratyphi A* (macrophage infections; see¹⁴). (*Shigella*: FDR<0.2, *Salmonella*: FDR<0.05, and $|\log_2FC|>0.5$).

8. Briefly discuss limitations of this study. What about genes not called in this work under these conditions? Are they not important?

Response: As specified above, the TraDIS screen presented here gives a snapshot of what happens up until 6h p.i. in the enteroid infection model. We have optimized the assay to strike a balance between a large enough intracellular population and minimizing the impact of epithelial cell death. We have now clarified the predominant study limitations, including the fact that spread and immune evasion mechanisms acting on longer time scales might not be efficiently picked up (Lines 336-337). Please also see our response to comments 5-6 by reviewer #1.

REVIEWER #2 (intestinal organoid co-cultures)

Di Martino, Jenniches et al. present a genome-wide screen of invasion factors in Shigella flexneri. Using an elegant combination of Transposon Directed Insertion Sequencing in S. flexneri and organoid-based invasion studies, the authors identify more than 100 genes linked to bacterial invasion into healthy human gut epithelial cells. The identification of 2 tRNA modification enzymes that regulate invasion-linked virulence gene expression represents an unexpected finding of this study that is convincingly linked to invasion modulation. Overall, this study is performed to a very high standard with numerous steps of technical establishment to ensure appropriate identification of stochastic mutant losses. I expect it to serve as a gold standard for similar future studies.

Response: We thank reviewer #2 for these positive remarks.

The main area where I see much untapped potential and some weaknesses is the comparative use of different organoid lines and protocols. The following points could substantially strengthen this manuscript:

1. If I interpret the methods section correctly, one jejunal and one colonic organoid line from different patients were used throughout this study. While the comparative statements between GI tract regions are highly interesting, the variation between these 2 patients is a striking confounding factor for any such statements in the current setup. Running the entire TraDIS screen in more organoid lines may be excessive, but validation of the key KO strains in at least 2 more organoid lines per intestinal segment (ideally paired from the same patient, but I understand that this may not be feasible) would be necessary to substantiate these statements.

2. In Figure 3 K/M, few patterns of apical-basal preference seem to be preserved. I have concerns about the robustness of this assay, these should be resolved by further replicates and ideally more organoid lines (see 1). The potential divergence (or neutrality) between apical and basal invasion is an intriguing aspect that can be studied very well with the platform developed by the authors, so that more robust statements on this aspect will further strengthen the manuscript.

Response to comment 1-2: The overall concern raised in comments 1-2 has to do with how robustly results from a given enteroid or colonoid line generalize across multiple independently established lines. This comment is highly relevant. As the reviewer points out, we found that there is no obvious difference in the *Shigella* effector requirement for invasion of jejunum- vs colon-derived human IECs, but that our results identified a context-dependent requirement for IpaA specifically during apical epithelial invasion (in both jejunal and colonic IECs). Still, it would be a strength to ensure that these findings reproduce across further donor cultures.

Along this reasoning, we have conducted additional barcoded consortium infections in additional enteroid and colonoid lines, and used the IpaA-mutant-containing consortium as the test case for reproducibility. Specifically, we have repeated barcoded infections in BO and AO enteroid and colonoid cultures derived from new tissue donors (Hu22-2jej and Hu22-4col) not previously included in the study. For each infection, an equally mixed (1:1:1:1:1:1) inoculum of six strains: two wt (tagA, tagB), two $\Delta mx i D$ (tagC, tagD) and two $\Delta ip a A$ (tagE, tagF) strains was prepared. We infected BO and AO enteroids and colonoids for 6h at MOI 40 (i.e. same conditions as in the earlier experiments), including 4 biological replicate consortia. The results of these experiments are included in FigS5 – new Panels c-f (see below). As before, analysis of the intracellular *Shigella* population revealed that the two non-invasive $\Delta mx i D$ control strains in each case were ~100-1000 fold less abundant than the two wt strains (Fig3j,l; FigS5c,e). Deletion of IpaA ($\Delta ip a A$) again had a minor impact in BO enteroids/colonoids (mutant ~30-50% less invasive than the wt strain), but caused a ~3.5-10-fold attenuation in AO enteroids/colonoids. This confirms that the context-dependent requirement for IpaA specifically during apical epithelial invasion is not dependent on patient-specific enteroid/colonoid characteristics. Moreover, the robust and internally controlled nature of barcoded infections is further highlighted by these experiments, as evident from the high degree of consistency between biological replicates and between experiments conducted in separate enteroid/colonoid lines (Fig3j,l; FigS5c,e).

FigS5 – Panels c-f. (c) BO and AO enteroids (ID 22-2jej) were infected with a mixed barcoded consortium comprising two wt (tagA, tagB), two $\Delta mxiD$ (tagC, tagD) and two $\Delta ipaA$ (tagE, tagF) strains at MOI 40 for 6h. The quantification of relative tag abundance, normalized against the corresponding input, was obtained by qPCR. For each geometry, we performed the infections with four independently generated consortia. Statistical significance determined by paired t-test for each specific mutant between the normalized output and the normalized input abundances (see Methods). * $p < 0.05$, ** $p < 0.01$, *** $p < 0.001$. (d) Bar graph showing the Colonization Index for each mutant in BO (dark grey) and AO (light grey) enteroid 22-2jej infections (derived from data in FigS5c and calculated as $1 - (wt/mut)$). wt is shown as 0. (e) BO and AO colonoids (ID 22-4col) were infected with a mixed barcoded consortium comprising two wt (tagA, tagB), two $\Delta mxiD$ (tagC, tagD) and two $\Delta ipaA$ (tagE, tagF) strains at MOI 40 for 6h. The quantification of relative tag abundance, normalized against the corresponding input, was obtained by qPCR. For each geometry, we performed the infections with four independently generated consortia. Gray shading indicates the detection limit. Statistical significance determined by paired t-test between the normalized output and the normalized input abundances (see Methods). * $p < 0.05$, ** $p < 0.01$, *** $p < 0.001$. (f) Bar graph showing the Colonization Index for each mutant in BO (dark grey) and AO (light grey) colonoid 22-4col infections (derived from data in FigS5e and calculated as $1 - (wt/mut)$). wt is shown as 0.

3. The Gentamicin protection assay used throughout this study may be validated in greater depth (or supported by further literature references). How were the concentrations of 200 $\mu\text{g/ml}$ for 2h or 20 $\mu\text{g/ml}$ for 6h p.i. determined, and are those sufficient to kill all bacteria of this strain in the absence of organoids cells?

Response: Gentamicin is an antibiotic with limited ability to penetrate host cells. However, the use of high concentrations of gentamicin over a prolonged time period should be avoided to prevent a potential exposure of intracellular bacteria to gentamicin due to pinocytosis. 200 $\mu\text{g/ml}$ gentamicin is a relatively high concentration and we therefore used it only for a short period of time (up to 2h) and then switched to a lower concentration of 20 $\mu\text{g/ml}$ for the remaining incubation time 3-6h p.i. As shown in Fig1c, this experimental setup ensures a robust resolution between the invasive wt *Shigella* strain and the $\Delta mxiD$ strain (non-invasive; lacking a structural component of the T3SS). This provides a wide dynamic range between fully invasive and essentially non-invasive strains. Nevertheless, we have now further confirmed the impact of these gentamicin concentrations on *Shigella*, as suggested by reviewer #2. Specifically, we have treated ~32 million pure *Shigella* wt bacteria (corresponding to the inoculum size in our infections assays) with 200 $\mu\text{g/ml}$ gentamicin for 2h in two different media, LB and Organoid Growth Media (OGM), to recapitulate the conditions encountered

during enteroid/colonoid infections. We then plated both untreated (input) and treated bacteria and counted the resulting CFUs. Exposure to 200µg/ml gentamicin for 2h at 37°C completely eradicated *Shigella* growth in both LB and OGM media (FigR4 – only shown in this response letter). This confirms that the gentamicin concentrations used in the enteroid/colonoid infections completely eradicate extracellular *Shigella* in liquid media. From this we can also conclude that in the enteroid infection setup, a small *Shigella* population makes it to a Gentamicin-protected state also in the absence of T3SS activity (see FigR1 above, $\Delta mxiD$ strain). However, that subpopulation represents only ~1% or less of the total enteroid/colonoid-colonizing *Shigella* population.

FigR4. *Shigella* wt CFU counts upon exposure to 200µg/ml gentamicin for 2h at 37°C in LB and OGM media. Input represents the initial population. Bacterial cultures were serially diluted and plated on LB agar plates. CFU counts come from 4 biological replicates.

Beyond these points, I think the study is exceptionally well-done, robust, interesting and warrants publication.

Response: We are happy to hear such a positive overall assessment of our work.

REVIEWER #3 (*Shigella*/host interactions)

*The manuscript by Di Martino and coworkers describes the application of TraDIS to identify genes relevant for *Shigella flexneri* infection of gut epithelial organoid models. The manuscript depicts the optimization of the organoids as a *Shigella* infection model amenable to the TraDIS large-scale approach. Through this approach, both chromosomally and plasmid-encoded genes that modulate the infection were identified, including a minimal set of T3SS effectors and two tRNA modifying enzymes.*

*Overall, this study describes an elegant methodology combining TraDIS and organoid models to tackle an important area of research, specifically *Shigella* infection. This is pertinent given the lack of relevant animal models to study infection by this bacterial pathogen. The manuscript results constitute a useful resource for the entire community.*

However, while the manuscript is interesting from a methodology and resource standpoint, the follow-up experiments on selected hits do not provide novel biology/mechanistic insights.

All the comments below are aimed at a clarification of the methodology and dataset.

Response: We thank reviewer #3 for the positive assessment of our approach and for constructive remarks, which have been considered as specified below. We would here like to emphasize that in addition to the technical framework and resource advances provided, this study also offers several novel insights into the biology of the infection process, for example: i) the discovery of >70 chromosomal *Shigella* genes not previously linked to an infection phenotype; ii) demonstration of which of the *Shigella* T3SS effectors are required for colonizing

non-transformed human small and large intestinal IECs from the apical versus the basal pole, and iii) elucidation of how tRNA modification enzymes and selective codon usage can be employed to globally govern *Shigella* virulence machinery expression for epithelial colonization.

1. Although I agree with the authors that there are severe limitations regarding animal models recapitulating Shigella infection in humans and that organoids are, in principle, more biologically relevant than cancer cell lines, the strong emphasis on this point should be toned down. Along this line, it would be highly relevant to discuss also the limitations inherent to the use of organoid models in the context of infection, as well of their applicability to high-throughput approaches.

Response: In response to this reviewer comment and similar suggestions by reviewer #1, we have clarified the limitations of the study and the applicability of high-throughput approaches to organoid models of infections (Lines 334-337). We have also carefully pruned the wording throughout to not overly emphasize the physiological significance of the enteroid/colonoid models.

2. The authors mention throughout the manuscript that the genes identified are essential for epithelial cell invasion. The use of the term invasion is rather surprising and not accurate, given that the TraDIS and most validation experiments were performed at 6 hpi. As such, the identified factors could affect (likely are) other steps of the interaction of the bacteria with epithelial cells (e.g. bacterial replication and/or clearance).

Response: Thank you for pointing out this important need for clarification. This comment aligns with similar requests from Reviewer #1. As elaborated on extensively above (Reviewer #1, comments 3-5), we have conducted additional experiments and analyses to pinpoint how *Shigella* invasion, intraepithelial replication and spread, and epithelial cell death, contribute within this “early colonization window” used in the TraDIS screen. This has led to the inclusion of a new FigS1 and the corresponding clarifications in the results text. Moreover, we have now consistently used the term “colonization” rather than “invasion” wherever this appears the more appropriate term.

3. Regarding the description of TraDIS, specifically the optimized protocol used for screening of the full library, the authors describe the application to the 43 sub-libraries as replicates of infection (Fig 2A and text). The use of replicates in this context does not appear accurate.

Response: We can see that this word choice was suboptimal. Our intention was to refer to the 43 enteroid infection experiments, performed with 43 different Tn5 *Shigella* sub-libraries. We agree that these should not be seen as “replicates” in the strict sense of the word (since each sub-library is unique), but rather as 43 reruns of an identical experimental protocol. We have now changed the phrasing to clarify this aspect (Lines 642-643).

4. Why was the infection/sequencing for the sub-library 11 not repeated?

Response: As specified in the methods section, we sequenced a pool of the 43 input and the 43 output libraries on a NextSeq 500 instrument over three runs using the 75-cycle High Output kit with a custom dark cycle recipe. In all three runs, the input sample for library 11 gave essentially no sequencing reads. Since we had a comparable amount of gDNA from this sample, this likely indicates that a problem during the sequencing library preparation occurred. Considering that a single sub-library only accounted for ~2-3,000 additional mutants, we decided to exclude this sample, since the very limited added benefit of resequencing would not justify the substantial time investment, logistics challenge, and added costs this would entail.

5. What is the overlap between the $\Delta mnmE/WT$ and $\Delta mnmG/WT$ differentially expressed proteins? Are the two tRNA modification enzymes functionally redundant? It would be interesting to add mechanistic and biological relevance for the data provided.

Response: We appreciate this useful request. As commented on in the results section, MnmE and MnmG work as a complex to modify the position 5 of the wobble uridine (U34) in the anticodon of specific tRNA species, which was previously dissected at the biochemical level (see^{15–17}). Therefore, it appears in line with current molecular biology knowledge that we found both genes to be similarly important for *Shigella* enteroid colonization. To emphasize this important point to the readers, we have added two new panels to FigS5 (now FigS6). The first shows the striking overlap between the $\Delta mnmE/WT$ and $\Delta mnmG/WT$ differentially abundant proteins (FigS6f). Moreover, a linear regression analysis between the $\Delta mnmE/WT$ and $\Delta mnmG/WT$ significantly differentially abundant proteins located on pINV further demonstrates the close agreement between the two independent data sets (FigS6g; below).

FigS6 – Panels f-g. (f) Venn-diagram of *Shigella* differentially expressed proteins in the following comparisons: $\Delta mnmE$ vs wt (MnmE regulated proteins), and $\Delta mnmG$ vs wt (MnmG regulated proteins); $\text{Log}_2\text{FC} \geq 0.5$, $\text{adj_pvalue} \leq 0.01$. (g) Graph showing linear regression between pINV differentially expressed proteins in the following comparisons: $\Delta mnmE$ vs wt, and $\Delta mnmG$ vs wt; $\text{Log}_2\text{FC} \geq 0.5$, $\text{adj_pvalue} \leq 0.01$.

We thank the reviewers for their constructive input on how to improve the manuscript. In addition to revisions instilled by the reviewer comments, we have also carefully pruned the manuscript for consistency and clarity, and ensured that the journal formatting requirements are met. We hope that this revised manuscript version will now be suitable for publication in Nature Genetics.

The Authors

REFERENCES

1. Di Martino, M. L., Falconi, M., Micheli, G., Colonna, B. & Prosseda, G. The multifaceted activity of the VirF regulatory protein in the *Shigella* Lifestyle. *Front. Mol. Biosci.* **3**, (2016).
2. Prosseda, G. *et al.* The virF promoter in *Shigella*: more than just a curved DNA stretch. *Mol. Microbiol.* **51**, 523–37 (2004).
3. Schroeder, G. N. & Hilbi, H. Molecular Pathogenesis of *Shigella* spp.: Controlling Host Cell Signaling, Invasion, and Death by Type III Secretion. *Clin. Microbiol. Rev.* **21**, 134–

- 156 (2008).
4. Cervantes-Rivera, R., Tronnet, S. & Puhar, A. Complete genome sequence and annotation of the laboratory reference strain *Shigella flexneri* serotype 5a M90T and genome-wide transcriptional start site determination. *BMC Genomics* **21**, (2020).
 5. West, N. P. *et al.* Optimization of virulence functions through glucosylation of *Shigella* LPS. *Science* (80-.). **307**, 1313–1317 (2005).
 6. Hong, M. & Payne, S. M. Effect of mutations in *Shigella flexneri* chromosomal and plasmid-encoded lipopolysaccharide genes on invasion and serum resistance. *Mol. Microbiol.* **24**, 779–791 (1997).
 7. Caboni, M. *et al.* An O Antigen Capsule Modulates Bacterial Pathogenesis in *Shigella sonnei*. *PLOS Pathog.* **11**, e1004749 (2015).
 8. Watson, J. L. *et al.* *Shigella sonnei* O-Antigen Inhibits Internalization, Vacuole Escape, and Inflammasome Activation. *MBio* **10**, (2019).
 9. Geiser, P. *et al.* Determinants of the Divergent *Salmonella* and *Shigella* Epithelial Colonization Strategies Resolved in Human Enteroids and Colonoids. *bioRxiv* 2024.05.03.592388 (2024). doi:10.1101/2024.05.03.592388
 10. Bourdet-Sicard, R. *et al.* Binding of the *Shigella* protein IpaA to vinculin induces F-actin depolymerization. *EMBO J.* **18**, 5853–5862 (1999).
 11. Tran Van Nhieu, G., Ben-Ze'ev, A. & Sansonetti, P. J. Modulation of bacterial entry into epithelial cells by association between vinculin and the *Shigella* IpaA invasin. *EMBO J.* **16**, 2717–2729 (1997).
 12. van Rijn, J. M. *et al.* High-Definition DIC Imaging Uncovers Transient Stages of Pathogen Infection Cycles on the Surface of Human Adult Stem Cell-Derived Intestinal Epithelium. *MBio* **13**, (2021).
 13. Rijn, J. M. van *et al.* Maturation of Human Intestinal Epithelial Cell Layers Fortifies the Apical Surface against *Salmonella* Attack. *bioRxiv* 2024.07.11.603014 (2024). doi:10.1101/2024.07.11.603014
 14. Wang, B. X. *et al.* High-throughput fitness experiments reveal specific vulnerabilities of human-adapted *Salmonella* during stress and infection. *Nat. Genet.* 2024 566 **56**, 1288–1299 (2024).
 15. Armengod, M. E. *et al.* Enzymology of tRNA modification in the bacterial MnmEG pathway. *Biochimie* **94**, 1510–1520 (2012).
 16. Armengod, M. E. *et al.* Modification of the wobble uridine in bacterial and mitochondrial tRNAs reading NNA/NNG triplets of 2-codon boxes. *RNA Biol.* **11**, 1495–1507 (2014).
 17. Moukadiri, I., Garzón, M. J., Björk, G. R. & Armengod, M. E. The output of the tRNA modification pathways controlled by the *Escherichia coli* MnmEG and MnmC enzymes depends on the growth conditions and the tRNA species. *Nucleic Acids Res.* **42**, 2602–2623 (2014).

Uppsala/Würzburg

April 8th, 2025

We thank the reviewers for the reassessment of our revised manuscript NG-A66137R. Below, we acknowledge their final comments on the revised manuscript (reviewer comments inserted verbatim as italicized text, followed by our replies).

Reviewer #1:

“This is an exciting time for Shigella infection biology. Novel, reliable infection models are desperately needed. In this revised manuscript authors carefully consider all reviewer comments, and fully address the breadth of reviews with additional experiments and text updates.

I have no further comments and look forward to the impact this work will have on the infection biology community.”

Author response: We appreciate the constructive input from reviewer #1 on the original manuscript version and are satisfied to hear that the revisions are deemed complete.

Reviewer #1’s feedback on response to Reviewer #2’s original comments:

“I continue to enjoy thinking about the manuscript and its impact.

Reviewer2 is positive: ‘...elegant combination...’, ‘...convincingly linked..’, ‘..study performed to a very high standard...’, ‘...serve as a gold standard...’....’Beyond these points, I think the study is exceptionally well done, robust, interesting and warrants publication.’

Reviewer2 comment 1, 2 (linked) about untapped potential / weakness: As acknowledged by the authors, this comment is highly relevant and results would be more convincing if reproduced across further donor cultures. To address this authors advance barcoded consortium infections in additional enteroid and colonoid lines. New results confirm context independent requirement for mxiD and context dependent requirement for lpaA. Importantly, consistency between biological replicates and between experiments conducted in separate enteroid / colonoid lines is compelling.

Reviewer2 comment 3 about gentamicin. Additional experiments were performed, clearly showing that gentamicin concentrations used in enteroid / colonoid infections eradicate extracellular Shigella.

Together, I feel authors have carefully considered and addressed comments from all reviewers (including reviewer2) and the manuscript is significantly strengthened as a result.”

Author response: Thank you for these enthusiastic remarks. With this, we conclude that also the original comments by reviewer #2 have been adequately addressed.

Reviewer #3:

“The authors have done extensive work and have introduced changes to the manuscript to address the main comments of all reviewers, and the revised version of the manuscript is significantly improved.

As mentioned previously, I find that this manuscript provides an important and comprehensive resource for researchers working on Shigella infections.”

Author response: We are grateful also to reviewer #3 for useful input on the original manuscript version and happily acknowledge that our revisions have been deemed complete.